# An experimental 2D-Var retrieval using AMSR2

David Ian Duncan[1,2], Patrick Eriksson[1], and Simon Pfreundschuh[1]

[1]Department of Earth, Space, and Environment, Chalmers University of Technology, SE 412 96 Gothenburg, Sweden
[2]*Now at European Centre for Medium-Range Weather Forecasts, RG2 9AX, Reading, United Kingdom

**Correspondence:** David Ian Duncan (david.duncan@ecmwf.int)

**Abstract.** A two-dimensional variational retrieval (2D-Var) is presented for a passive microwave imager. The overlapping antenna patterns of all frequencies from the Advanced Microwave Scanning Radiometer-2 (AMSR2) are explicitly simulated to attempt retrieval of near surface wind speed and surface skin temperature at finer spatial scales than individual antenna beams. This is achieved, with the effective spatial resolution of retrieved parameters judged by analysis of 2D-Var averaging kernels. Sea surface temperature retrievals achieve about $30\,\mathrm{km}$ resolution, with wind speed retrievals at about $10\,\mathrm{km}$ resolution. It is argued that multi-dimensional optimal estimation permits greater use of total information content from microwave sensors than other methods, with no compromises on target resolution needed; instead, various targets are retrieved at the highest possible spatial resolution, driven by the channels' sensitivities. All AMSR2 channels can be simulated within near their published noise characteristics for observed clear-sky scenes, though calibration and emissivity model errors are key challenges. This experimental retrieval shows the feasibility of 2D-Var for cloud-free retrievals, and opens the possibility of standalone 3D-Var retrievals of water vapour and hydrometeor fields from microwave imagers in the future. The results have implications for future satellite missions and sensor design, as spatial oversampling can somewhat mitigate the need for larger antennas in the push for higher spatial resolution.

## 1 Introduction

Observations from satellites at microwave frequencies for which the Earth's atmosphere is relatively transparent, so-called imaging channels, provide valuable information to constrain estimates of skin temperature, wind speed, sea ice concentrations, soil moisture, and more. With global coverage, relative insensitivity to clouds, and a decades-long data record, microwave radiometers provide complementary information alongside infrared and scatterometer retrievals that may possess higher accuracy or higher resolution but have limitations in coverage (Atlas et al., 2011; Reynolds et al., 2007). Additionally, due to microwave imager channels' multiple sensitivities, simultaneous retrieval of multiple parameters with one observation vector ensures some geophysical consistency in the retrieved state vector (e.g. Phalippou, 1996; Bettenhausen et al., 2006; Munchak et al., 2016; Duncan et al., 2018). In operational contexts, assimilation of microwave imager radiances sensitive to skin temperature, humidity, and winds can add to forecast skill in various ways (Geer et al., 2017; Singh et al., 2008; Kazumori et al., 2016).

Lower frequency microwaves have better sensitivity to the Earth's surface, in this context constituting frequencies of about $10\,\mathrm{GHz}$ and below, with wavelengths of about $3\,\mathrm{cm}$ and more, known also as the X- and C-bands. These bands are valuable for

Earth observation because the atmosphere attenuates these signals very little, with thermal emission from the Earth's surface impeded only slightly by water vapor and precipitation. Apart from radio frequency interference (RFI) being common at these frequencies due to the utility of non-attenuating bands for telecommunications (Draper, 2018; Zabolotskikh et al., 2015), the main drawback of these frequencies for Earth observation is their spatial resolution, which is driven by laws of diffraction. The spatial extent of one satellite radiance measurement is determined by the observation frequency, orbit altitude, and antenna size. With the satellite's altitude fixed, the spatial resolution is linearly proportional to the ratio of wavelength to reflector diameter.

The largest passive microwave reflectors yet flown for Earth observation include the JAXA Global Change Observation Mission - Water (GCOM-W; Imaoka et al. (2010)) with the Advanced Microwave Scanning Radiometer 2 (AMSR2), which features a $2.0\,\mathrm{m}$ solid reflector, and NASA's Soil Moisture Active Passive (SMAP) satellite with a $6\,\mathrm{m}$ deployable reflector (Entekhabi et al., 2010). Both of these have real aperture antennas; because large solid reflectors cause concerns over their mass, drag, and transport into orbit, other antenna types are being explored (Kilic et al., 2018). But even with larger antennas, the trade off between sensor size and resolution on the ground remains.

However, a hidden advantage of observation at these bands lies in the spatial oversampling that is a byproduct of the large field of view (FOV) of each channel. Essentially, because the footprints are so large, they overlap significantly both along individual scans and between scans. This characteristic can be exploited by averaging or smoothing as a way to beat down the sensor noise, such as by averaging each channel's radiometric signal on an equal area grid before running a retrieval (Meier et al., 2017). But this further smears the low resolution of the large FOVs and discards any finer-scale information that may exist in the measurements.

Spatial oversampling of satellite radiometry should permit retrieved quantities to achieve greater robustness and a higher effective spatial resolution. As with photographic images, satellite imagery can benefit from techniques that allow finer spatial scales to be realised and crisper, more detailed images to emerge. However, as wavelength increases, physical constraints limit the design of radiometers, necessitating ever larger antennas to achieve finer spatial resolution (e.g. Kilic et al., 2018; Entekhabi et al., 2010). This is a design constraint when observing geophysical properties best sensed via low frequency microwaves. This has been an issue for past satellite sensors and will remain a challenge for maximal use of future sensors such as the MicroWave Imager (MWI) and Ice Cloud Imager (ICI) on MetOp-SG, which will also be heavily over-sampled.

To put the spatial oversampling of microwave imagers into context, it is worth examining the lowest frequency channels of AMSR2. The $6.925\,\mathrm{GHz}$ channels on AMSR2 observe the Earth's surface with a half-power beam width (HPBW) of $35\,\mathrm{km}$ by $62\,\mathrm{km}$ in the across- and along-track directions, respectively, defining their instantaneous FOV (IFOV). The AMSR2 produces 243 observation centres across one scan, spanning about $1450\,\mathrm{km}$, and approximately 4000 scans per orbit. The average scans are $10\,\mathrm{km}$ apart along-track, with the average observations centres just $9\,\mathrm{km}$ apart across-track at swath centre and closer at the swath's edge. To treat adjacent measurements at low frequency channels as having wholly independent information is demonstrably problematic, as a large percentage of the information is redundant. The relative overlap of AMSR2 beams on the Earth's surface and the large variations of FOV size with frequency can be seen in Fig. 1.

For some retrieval targets such as precipitation, there is little spatial coherence to an image, or spatial correlations are highly variable. But for a retrieval target like sea surface temperature (SST) or ocean wind speeds, a priori knowledge of its spatial

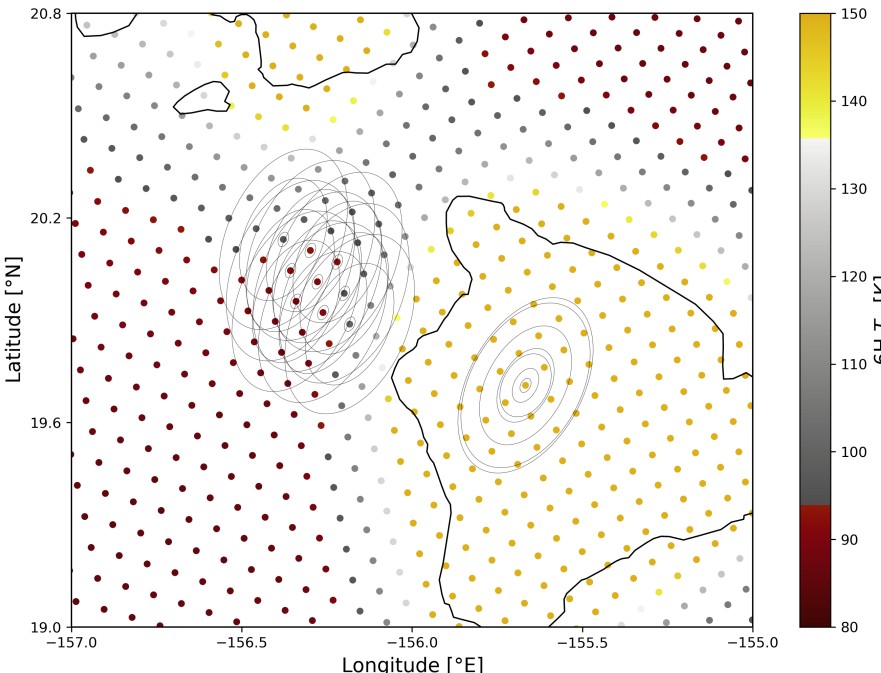

**Figure 1.** An example of FOV sizes and measured $T_B$ at the 6H channel near the Island of Hawai'i, from an AMSR2 orbit on September 21, 2016. The $T_B$s at 6H are shown in coloured dots, with the HPBW given at all AMSR2 frequencies for one observation centre viewing the island itself. The HPBW is outlined for three frequencies only (6, 10, 89GHz), for three consecutive observation centres along- and across-scan just west of the island, so as to view the spatial overlap without too much clutter.

structures can be brought to bear on the inversion problem as a further constraint. Sub-mesoscale fluctuations in SST fields can exist (Castro et al., 2017), but most oceanic near-surface wind speed variability is synoptically forced and thus spatial decorrelation lengths are significant. Wind retrievals can also benefit from accounting for spatial correlations for ambiguity removal (Vogelzang and Stoffelen, 2018; Lin et al., 2016). Ideally, spatial covariances of each field can be included in a retrieval

5   scheme, and even cross-correlations between variables. The common practice of treating neighboring satellite measurements as wholly independent retrieval targets thus represents a suboptimal use of the total information content of the observation vector, especially considering the substantially overlapping FOVs of adjacent observations (Fig. 1).

    The dependence of FOV size on frequency has led to convolution and de-convolution approaches (e.g. Backus and Gilbert, 1967; Petty and Bennartz, 2017) to construct a common spatial resolution for performing retrievals. This can be crucial for

10   1D-Var retrievals with scene inhomogeneities (Rapp et al., 2009). However, the optimal observation resolution depends upon the retrieval target, may vary over the scene, and is not known a priori. The choice of a common spatial resolution is a compromise and adds noise to some of the observed quantities. In contrast, a 2D-Var approach permits usage of all native resolution observations with no compromises required. If the various beam sizes, angles, positions, and antenna patterns are accounted for in the forward model, and spatial homogeneity within the beam is not required because the forward model

contains horizontal dimensions, then it is possible to rely on fewer assumptions. If the beams are adequately simulated and the retrieval grid is finer than the beam widths, 2D-Var (or 3D-Var) thus offers an opportunity to use the native resolution observations without resampling or compromise, a maximal usage of the available information content.

One hallmark of data assimilation is the inclusion of many variables and their covariances as part of a multidimensional minimisation procedure to determine the geophysical state. A purely variational assimilation in the three spatial dimensions is known as 3D-Var, or 4D-Var if a time component is included as well. Standalone remote sensing retrievals also endeavour to solve for the geophysical state but typically have a much more limited scope, both spatially and in terms of the state vector. Satellite retrievals may use any method to invert the measurement vector, but retrievals are usually treated independently in space and time, with no knowledge of adjacent measurements. In other words, each observation that comprises the image is treated separately. So while 1D-Var retrieval methods are common for satellite retrievals (e.g. Boukabara et al., 2011; Elsaesser and Kummerow, 2008; Duncan and Kummerow, 2016), the horizontal spatial dimensions are usually disregarded for downward-viewing sensors. As weather models progress to spatial resolutions finer than some satellite FOVs, the need for better use of satellite sensors' inherent spatial information will only increase. Furthermore, as full utilisation of sensors' information is a goal of data assimilation, any approach that can make better use of the spatial information contained in satellite observations has potential benefits for future data assimilation efforts.

In the following section, the theory behind 2D-Var and its application in this study are given. The sensor and forward model are then described in detail, as are the covariances and a priori information used in the 2D-Var inversion. To test out the retrieval methodology, synthetic retrieval results are attempted first and presented in Section 4. To complete the proof of concept, the 2D-Var inversion is then tested for two scenes with real observations from AMSR2 and presented in Section 5. Discussion and conclusions close the paper.

## 2  2D-Var

### 2.1  Theory

The optimal estimation methodology is a Bayesian inversion that balances prior knowledge and its known errors with the information gained by measurements with their own errors (Rodgers, 2000). Via variational iteration, optimal estimation attempts to minimise the following cost function that balances the a priori state vector ($x_a$) and the measurement vector ($y$) to invert the measurement with a forward model operator ($f(x)$) and determine the state vector ($x$) of maximum posterior likelihood:

$$\Phi = (y - f(x))^T \mathbf{S_y}^{-1}(y - f(x)) + (x - x_a)^T \mathbf{S_a}^{-1}(x - x_a) \tag{1}$$

The observation operator covariance matrix ($\mathbf{S_y}$) and a priori covariance matrix ($\mathbf{S_a}$) determine the trust placed in the measurement vector and a priori knowledge. In the context of passive microwave satellite retrievals, these are typically 1-dimensional variational retrievals, or 1D-Var (Boukabara et al., 2011; Duncan and Kummerow, 2016; Pearson et al., 2018), as the state vector exists in one spatial dimension. This cost function (Eq. 1) is applicable for inverse problems containing

zero to $n$ dimensions, and can thus be adapted from a 1D-Var retrieval to handling retrieval targets of multiple variables in $n$-dimensional space. Gaussian-distributed errors are assumed in this methodology.

An important byproduct of optimal estimation is its estimation of the posterior probability density function (PDF) of the retrieved state, providing users with a representation of the errors and covariances of retrieved parameters. This is described by $\mathbf{S_x}$,

$$\mathbf{S_x} = (\mathbf{K}^T \mathbf{S_y}^{-1} \mathbf{K} + \mathbf{S_a}^{-1})^{-1} \tag{2}$$

where $\mathbf{K}$ is the Jacobian defined by the first derivative of $f(x)$ to changes in the state vector. A retrieved parameter's "posterior error" is defined as the square root of that element from the diagonal of $\mathbf{S_x}$, actually signifying the width of the posterior PDF.

The averaging kernel matrix $\mathbf{A}$ is defined in relation to the Jacobian $\mathbf{K}$ and the covariance matrices $\mathbf{S_y}$ and $\mathbf{S_a}$, with its interpretation being the sensitivity of the retrieved state ($\hat{x}$) to the true state.

$$\mathbf{A} = (\mathbf{K}^T \mathbf{S_y}^{-1} \mathbf{K} + \mathbf{S_a}^{-1})^{-1} \mathbf{K}^T \mathbf{S_y}^{-1} \mathbf{K} = \frac{\delta \hat{x}}{\delta x} \tag{3}$$

The averaging kernel also helps to define the smoothing error of the retrieval, $\mathbf{S_s}$, in which $\mathbf{I}$ is the identity matrix.

$$\mathbf{S_s} = (\mathbf{A} - \mathbf{I}) \mathbf{S_a} (\mathbf{A} - \mathbf{I}) \tag{4}$$

Rows of $\mathbf{A}$ will sum to 1.0 (along the respective block of the sparse matrix) for retrieved variables for which the retrieval has full sensitivity to the true state (Rodgers, 2000). This is sometimes referred to as the "measurement response" (Baron et al., 2002), and will differ for each retrieved parameter at all retrieval grid points. Later in the study, rows of $\mathbf{A}$ are translated into the 2D space of the retrieval grid and examined as the measurement response for each retrieved parameter, indicating the spatial resolution achieved by the inversion for these retrieval targets. The trace of $\mathbf{A}$ defines the degrees of freedom for signal (DFS), or the total number of retrieved variables fully constrained by the retrieval. It is worth noting that the obtained $\mathbf{A}$ and $\mathbf{S_x}$ are both dependent upon the assumed covariances of the a priori state, $\mathbf{S_a}$, and observation operator, $\mathbf{S_y}$, and thus their interpretation relies on realistic user inputs for the elements in these matrices.

## 2.2 Retrieval setup

Following the terms presented in Eq. 1, here are detailed the covariances and a priori information used within the 2D-Var inversion. The a priori state ($x_a$) for SST and $10\,\mathrm{m}$ wind speed (hereafter WSP) in the synthetic case are defined simply as the field mean for all points in the retrieval grid; in the real cases, $x_a$ comes from reanalysis data interpolated to the retrieval grid. For all cases, the first guess values are the means of the $x_a$ fields.

A priori covariances ($\mathbf{S_a}$) are defined by a non-diagonal matrix, as non-zero spatial correlations are assumed for the parameters. For the cases presented, no cross correlation is prescribed between SST and WSP. The diagonal terms in $\mathbf{S_a}$ are fixed for both SST and WSP, indicating assumed standard deviations of $1.5\,\mathrm{K}$ and $1.5\,\mathrm{m\,s^{-1}}$. Off-diagonal terms in $\mathbf{S_a}$ are calculated via a decorrelation distance ($\ell_d$) and the proximity of each retrieval grid point in space, with the distance between grid points $i$ and $j$ given by $d_{i,j}$.

$$\mathbf{S_a}(i,j) = \sigma_a^{\,2} \exp(-\ell_d/d_{i,j}) \tag{5}$$

In all retrieval cases described, $\ell_d$ is prescribed as $1.0°$ in latitude and longitude. The retrieval grid spacing is set independently, and can be set separately for different retrieved parameters. For simplicity, the retrieval grid for both SST and WSP is set at $0.05°$ in latitude and longitude. This is finer than the spacing of most AMSR2 observation centres on the Earth's surface and indeed most global weather models, but in line with grid resolutions applied for scatterometers (Vogelzang and Stoffelen, 2017). If a near-zero decorrelation length were used, a coarser retrieval grid would be needed.

Observation error covariances ($\mathbf{S_y}$) are quite simple for this retrieval, with no off-diagonal terms considered. The matrix is diagonal, consisting of the published sensor noise characteristics for every channel (Table 1). This is increased by a factor of 2.0 for the real world retrievals to somewhat compensate for the existence of forward model and parameter errors. The diagonal $\mathbf{S_y}$ makes the inversion simpler and faster. For example, since an inversion of 14 channels for an observation space of 20 scans with 25 observation centres each constitutes an observation vector ($y$) of 7000 elements, this is not trivial. Whereas some studies include covariances between co-registered observations due to correlated forward model errors (Weston and Bormann, 2018; Duncan et al., 2018), it is unclear whether sensor noise or other errors have any spatial correlations. The nature of sensor noise and its assumed characteristics from one measurement to another, i.e. Gaussian and random, will be a topic raised again in Section 7.

## 3 Data and Methods

### 3.1 GCOM-W AMSR2

The AMSR2 sensor flies on JAXA's GCOM-W satellite, which was launched in May 2012. Due to its numerous low frequency microwave imaging channels sensitive to the surface, it is the sensor of interest for this study. The channels available on AMSR2 and their characteristics are given in Table 1. GCOM-W flies in a sun-synchronous low Earth orbit of inclination $98.2°$ with ascending node crossing the equator at 13:30 local time, part of the A-Train constellation, at a nominal altitude of $700\,\mathrm{km}$. AMSR2 is a conically scanning radiometer, maintaining a near-constant Earth incidence angle of $55°$ via a sensor angle of $47.5°$ off nadir.

Two main factors for retrieval performance with AMSR2 are its channels' noise characteristics, given as noise equivalent differential temperature (NEDT), and its instantaneous field of view (IFOV), given in kilometres and indicative of its HPBW as defined by an ellipse on the Earth's surface. These two characteristics change for each channel, affecting the precision and spatial extent of each individual observation's radiometric information. The sensor noise and FOV ultimately limit the accuracy of retrieval quantities, which for a variational retrieval can be investigated in terms of its posterior uncertainty and spatial smoothing error (Eq. 4).

All observational data used from AMSR2 come from JAXA's L1R product (Maeda et al., 2016). This data product contains resampled brightness temperatures ($T_B$s) at various channels, but only the native resolution $T_B$s are used in this study. In addition, the spacecraft's location and the latitude/longitude pair for each AMSR2 bore sight location on Earth are used to explicitly define the sensor line of sight through Earth's atmosphere.

**Table 1.** Sensor characteristics of AMSR2 aboard JAXA's GCOM-W satellite. Vertically and horizontally polarised channels are notated by V and H, respectively. Specification NEDT values are shown (Imaoka et al., 2010). Note that additional 89GHz channels exist on AMSR2 but are not included as they were not used here.

| Centre Frequency [GHz] | 6.925 | 7.3 | 10.65 | 18.7 | 23.8 | 36.5 | 89.0 |
|---|---|---|---|---|---|---|---|
| Polarisations | V, H | V, H | V, H | V, H | V, H | V, H | V, H |
| Bandwidth [MHz] | 350 | 350 | 100 | 200 | 400 | 1000 | 3000 |
| NEDT [K] | 0.34 | 0.43 | 0.70 | 0.70 | 0.60 | 0.70 | 1.20 |
| Angular beam width [°] | 1.80 | 1.80 | 1.20 | 0.65 | 0.75 | 0.35 | 0.15 |
| IFOV width [km] | 35 | 34 | 24 | 14 | 15 | 7 | 3 |
| IFOV height [km] | 62 | 58 | 42 | 22 | 26 | 12 | 5 |

Sensor calibration and its relative agreement with the forward model are of paramount importance for the success of a retrieval. In the case of SST retrieval, for example, the ocean surface emissivity is on order 0.55 and 0.25 for the 6V and 6H channels at 55° Earth incidence angle, respectively, and thus a systematic calibration bias of only $0.5\,\mathrm{K}$ could translate to a $1.0\,\mathrm{K}$ SST bias. Due to previous success using the intercalibration tables from the Global Precipitation Measurement (GPM)

mission (Berg et al., 2016) with a similar forward model (Duncan and Kummerow, 2016; Duncan et al., 2017), the AMSR2 channels used in the GPM constellation ($10\,\mathrm{GHz}$ and up) in this study use the GPM calibration (Berg, 2016). To make the 6V/6H/7V/7H channels consistent with this calibration, ad hoc adjustments were calculated and applied to return zero net bias at these channels over several orbits forced with reanalysis data and using the retrieval's forward model. The 6V and 7V calibration offsets applied are -0.46K and -0.63K, versus -2.59K and -3.12K for 6H and 7H. However, the calibration used here

is far from definitive due to the non-linear calibration adjustments inherent in the L1R data (which are largest for the lowest frequency channels), the ad hoc nature of the adjustments performed, and the limited scope of this analysis. A different forward model would require different calibration adjustments.

## 3.2 Forward model

As the forward model of the 2D-Var retrieval presented here is novel and key to its understanding, it is explained in detail.

The forward model and the 2D-Var solver itself are both defined within the Atmospheric Radiative Transfer Simulator (ARTS) V2.3 (Eriksson et al., 2011; Buehler et al., 2018); ARTS permits 3-dimensional polarised radiative transfer, which is necessary for the forward model. All code to call the ARTS model, set up the inversion, and analyze the retrieval output were written in Jupyter Notebooks using the Python package Typhon v0.7.0 (Typhon Authors, 2019), and are publicly available (Duncan, 2019). In this section and throughout the paper, "scan" refers to each set of observations swept out by the radiometer in one

rotation, throughout which spacecraft position is assumed constant; "pixel" refers to points within a scan, which is a vector of $T_B$s observed simultaneously and centred at one point on Earth's surface but with different antenna patterns. Thus pixels are points in the across-track direction, with numerous pixels per scan, while successive scans are in the along-track direction

relative to the spacecraft's motion vector. For reference, the AMSR2 L1R data contain 243 (low frequency) pixels per scan and thus the sample cases presented here represent a fraction of the total swath width.

A geophysical space is defined on a regular latitude and longitude grid, with a pressure grid defining the atmosphere above. For the synthetic cases in this study, a typical tropical atmospheric profile is assumed for temperature and water vapour; for the real observations, atmospheric profiles come from the European Centre for Medium-Range Weather Forecasts (ECMWF) Reanalysis 5 (ERA5) (Copernicus Climate Change Service (C3S), 2017). The atmospheric grid matches the pressure levels from ERA5, from sea level up to $100\,\text{hPa}$. Within the geophysical space, an independent retrieval grid is defined with its own spatial resolution and edges. In the examples given, all observations lie within the confines of the retrieval grid, and the retrieval grid is entirely within the larger geophysical grid. The term "observation space" will refer to the area of the retrieval grid well populated by observations.

Each AMSR2 scan is treated as a measurement block in ARTS, connecting each individual pixel. The measurement block is then described in angular space as deviations from the centre bore sight of the scan, in zenith (along-track) and azimuth (across-track). Sampling of this measurement space is accomplished by performing numerous "pencil beam" radiative transfer calculations on a defined angular grid in the 3D space. This angular grid's resolution is chosen based on the density of sampling desired of the antenna patterns and geophysical space. The pencil beam grid has a prescribed number of points between each bore sight, with this angular resolution also used in the zenith direction to keep consistent sampling between scans. For example, with 7 angular grid points between bore sights (both along- and across-track), a measurement block of 23 pixels across one scan can contain an angular grid of 201 by 25 pencil beams; in this example each scan is simulated using about 5000 pencil beams.

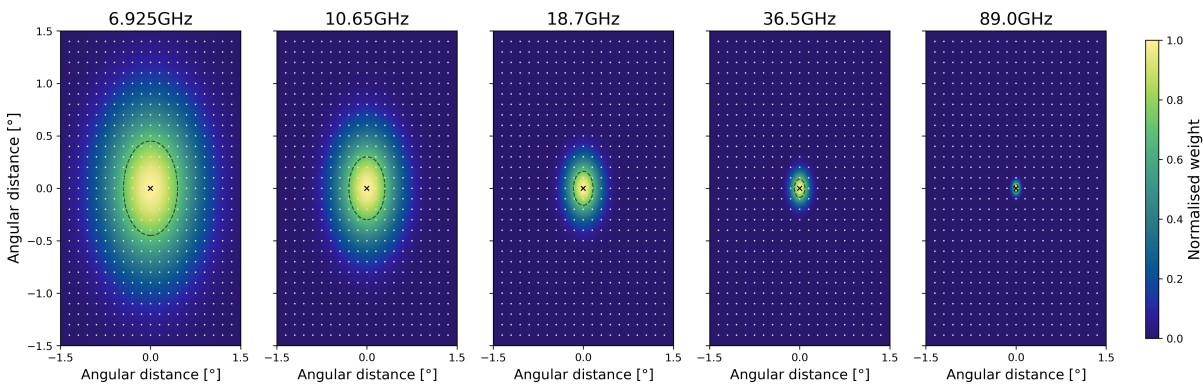

**Figure 2.** Antenna response in angular space for five modelled AMSR2 frequencies. The response is normalised to have a maximum value of 1.0 at the bore sight, defined by an 'x' in each panel, with angular distances defined in reference to the bore sight. The HPBW is shown by a dashed line in each panel, as defined in Table 1. The aspect ratio of the plots approximates the perspective if viewed at a $55°$ incidence angle on the Earth's surface, hence the elliptical shape. Grey dots signify a typical angular grid spacing for pencil beam calculations.

Weighting of these pencil beam calculations to synthesise the sum total $T_B$ response of the sensor is determined by the 2D antenna response, shown in Fig. 2 for five of the AMSR2 frequencies. Note that these are defined in angular space, as per the published angular beam width of each channel (Imaoka et al., 2010), assuming Gaussian response that is independent of channel polarisation. The assumption of Gaussian response with width defined by HPBW is a simplification (Maeda et al., 2016, Fig. 6). Non-negligible radiative response occurs outside the HPBW designated by ellipses in each panel, indicating that even the FOVs seen in Fig 1 can downplay the degree of overlap for AMSR2's low frequency observations. As a final step in the forward model, ARTS combines the measurement blocks of each scan to create one measurement vector, containing $T_B$s for each channel at every bore sight for the desired pixels and scans. An example of the pencil beam sampling of the antenna response patterns is visible in Fig. 2 by grey dots populating the angular grid. Because of the ARTS measurement block system, each pencil beam is used for weighting various bore sights' channels and thus effectively reused, reducing overall computational cost.

The ocean surface emissivity model employed in the forward model is FASTEM-6, the FAST microwave Emissivity Model version 6 (Kazumori and English, 2015). FASTEM calculates sea surface emissivity and reflectivity at microwave frequencies for vertical and horizontal polarisations, taking SST, WSP, sea surface salinity, wind direction, and atmospheric transmittance as inputs. In the cases given here, salinity is prescribed as 34psu, with wind direction and transmittance either derived from ERA5 input data or prescribed in the synthetic cases. It is noted that wind direction can impact observed $T_B$ by about 2 K at typical wind speeds (Meissner and Wentz, 2012; Kazumori and English, 2015) and its prescription is thus a potential source of error in the observational retrieval cases.

All the retrievals described utilise 12 AMSR2 channels, forgoing the 23V and 23H channels but using all others given in Table 1. The primary sensitivity of 23V and 23H is to column water vapour because of their proximity to the 22.235 GHz absorption line, and water vapour is not retrieved in the inversion. Their inclusion did not adversely impact the retrieval of surface parameters, but because analysis of these channels is not instructive to retrieval behaviour they were left out.

### 3.3 Optimal estimation module

A generic solver for inverse problems based on the optimal estimation framework (Rodgers, 2000) has been integrated into ARTS v2.3, effectively extending its functionality from a radiative transfer model to a retrieval framework. From the specification of retrieval quantities and a priori assumptions, the optimal estimation solver minimises the cost function (Eq. 1) using either a Gauss-Newton or Levenberg-Marquardt optimisation method, thereby iteratively recomputing simulated observations ($f(x)$) and Jacobians ($\mathbf{K}$) as the state vector is updated. Additional built-in functionality is provided to compute diagnostic quantities such as the a posteriori covariance matrix ($\mathbf{S}_x$) and averaging kernel matrix ($\mathbf{A}$).

### 4 Synthetic scene results

Before testing the 2D-Var methodology on real observations it was tested on a synthetic ocean scene. Synthetic observations of AMSR2 channels are first simulated for the "true" scene using the antenna patterns and sampling of Fig. 2, then random

Gaussian noise is added to the synthetic measurements according to the NEDT values from Table 1. This defines the observation vector, $y$. The measurement covariances ($\mathbf{S_y}$) match the noise added to $y$. An advantage of testing the 2D-Var on a synthetic scene is that the true state can be constructed to match exactly the standard deviations and spatial correlations prescribed in $\mathbf{S_a}$. This ensures that the posterior errors ($S_x$) and averaging kernel ($\mathbf{A}$) output by the synthetic retrieval are exactly correct, whereas real retrievals always contain caveats due to incomplete knowledge about the true $\mathbf{S_a}$ and $\mathbf{S_y}$. This is key to later conclusions, as $\mathbf{S_x}$ and $\mathbf{A}$ are exactly correct in the synthetic case by construction. These retrievals all use 12 channels and an observation area of 11 scans with 15 pixels from the middle of each scan.

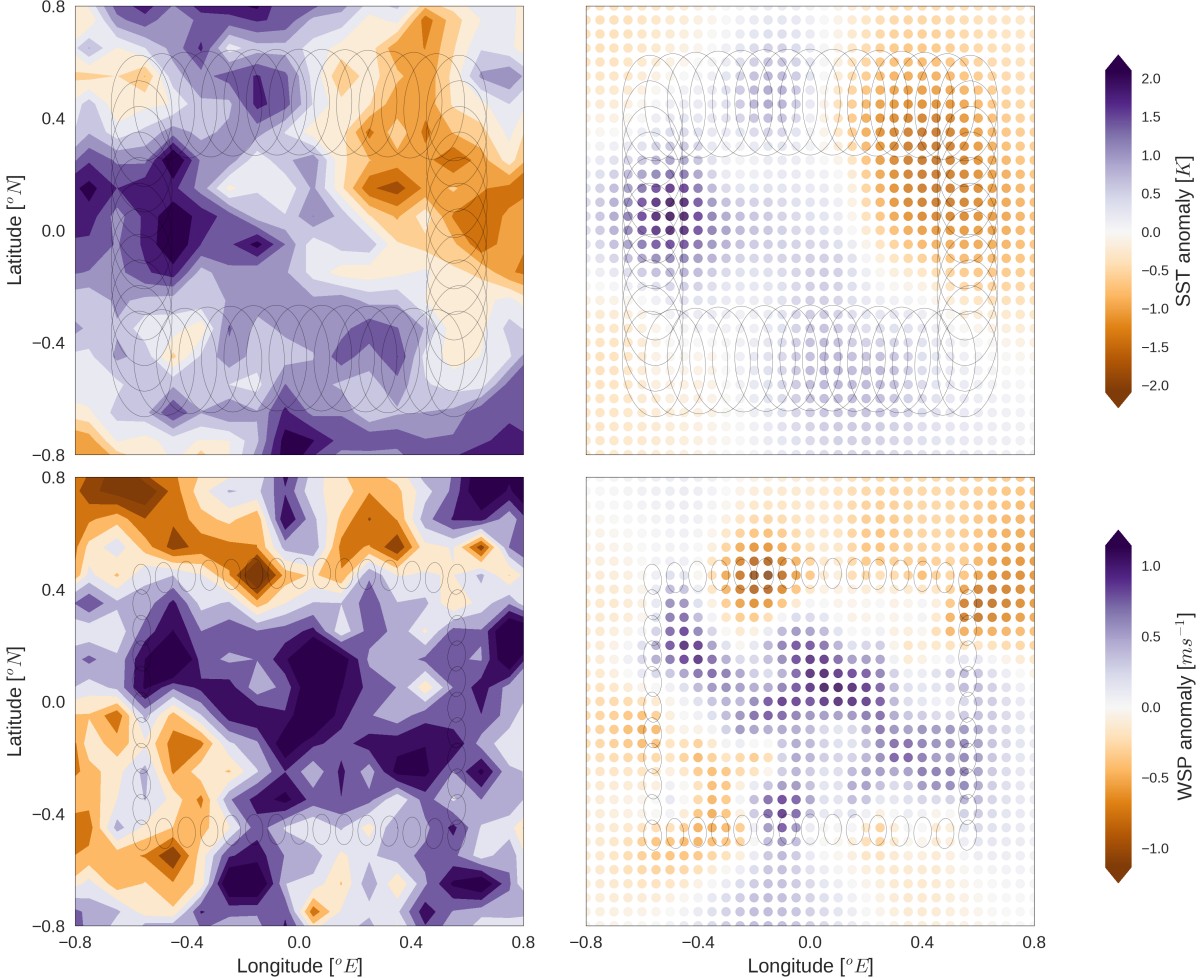

**Figure 3.** For the synthetic retrieval scene shown in contours (left panels), the retrieved state vectors are shown for SST (right top) and WSP (right bottom). Each dot represents a retrieval grid point. The fields are displayed as the anomaly from each field's mean. FOVs for the 6 and 36 GHz channels are shown for the pixels defining the edge of the observation area.

The synthetic retrieval case described is for a priori standard deviations of $1.5\,\mathrm{K}$ and $1.5\,\mathrm{m\,s^{-1}}$ and spatial decorrelation coefficients of $1.0°$, with the mean SST and WSP $292\,\mathrm{K}$ and $6.3\,\mathrm{m\,s^{-1}}$. Fig. 3 shows retrieved SST and WSP versus the true values. The 6V FOVs for the outermost pixels are shown to circumscribe the observation area within the retrieval grid, which has a $0.05°$ resolution as indicated by the dots.

5      Deviations between the true and retrieved state vectors in Fig. 3 show finer spatial variability in retrieved WSP than SST despite the same prescribed covariances. This is because the random sensor noise influences finer spatial scales for WSP retrieval, as the dominant information content for WSP comes from higher frequency channels with smaller FOVs. In contrast, the retrieved SST field is comparatively smooth relative to its true state. For both retrieval targets, deviations from the true state maximise outside the observation area signified by the ellipses, because there is little observational information to guide the

10      retrieval and thus the spatial correlations to the a priori state are dominant.

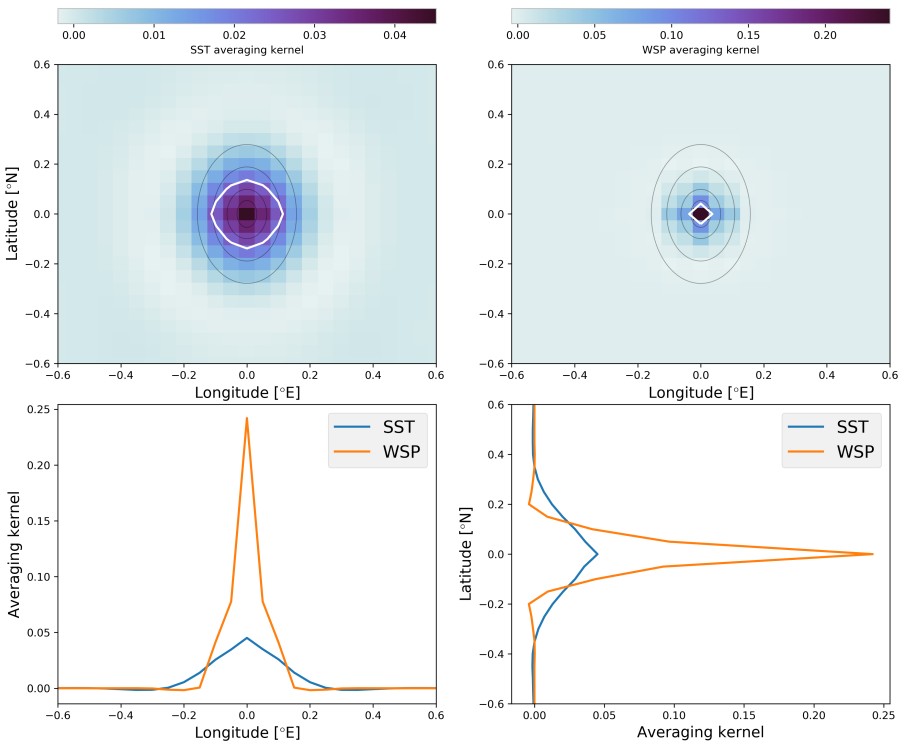

**Figure 4.** Averaging kernels for a central grid point from the synthetic scene, shown for SST (top left) and WSP (top right), with cross sections through the maxima given in the bottom panels. Overlaid on the top panels are the FOVs for 6, 10, 18, and 36GHz channels (in black) along with a white line indicating the half power contour of the averaging kernel.

From this same scene, the averaging kernels of a point near the middle of the retrieval grid are shown in Fig. 4. As with the definition of FOV, the white contour indicates where the averaging kernel drops to half of its peak value. The FOVs of the first four frequencies from Fig. 2 are overlaid in black. The bottom panels provide longitudinal and latitudinal slices through the

2D averaging kernel to demonstrate how peaked these fields are. Fig. 4 shows that the spatial resolution achieved for SST is approximately circular and lies mostly within the 10GHz FOV, whereas for WSP it has a much tighter achieved resolution that is on order of the retrieval grid itself. The sum of both averaging kernels is approximately 1.0, indicating that retrieval of both variables is fully constrained by the observations.

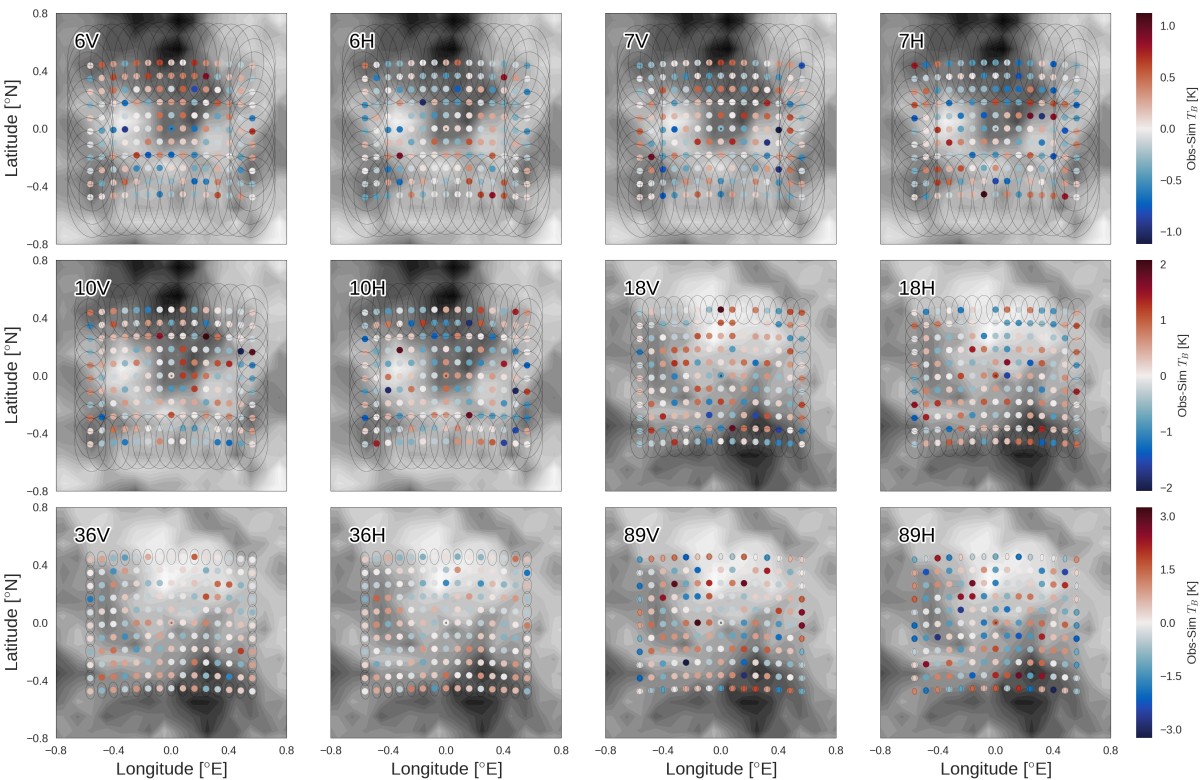

**Figure 5.** Observed minus simulated $T_B$ for the 12 AMSR2 channels used in the 2D-Var for the synthetic scene. Channel names are given in each panel. Backgrounds of each panel show contours of the SST (first six panels) and WSP fields (last six panels), echoing Fig. 3. FOVs for each channel are shown in black as ellipses for edge pixels.

5   The companion Fig. 5 gives the difference between synthetic "observed" $T_B$ and simulated $T_B$ at all channels for the same retrieval case. In other words, it compares the resultant forward modelled $T_B$ vector reached once the 2D-Var has converged on a solution ($f(x)$ in Eq. 1) with the observation vector ($y$). Spacing of the dots in Fig. 5 demonstrates the proximity of all the beam centres. The impact of the prescribed noise on the final fit with forward modelled $T_B$s is visible, manifest as a noisy field with standard deviation approximating NEDT for each channel. In the background of the panels the SST and WSP fields

10   are contoured as in Fig. 3, corresponding with the primary sensitivity of each channel, as the lower frequencies primarily sense SST and higher frequencies WSP. No systematic patterns are seen in the $T_B$ differences, corresponding neither between the channels nor to the state vector differences in Fig. 3.

The choice of retrieval grid resolution influences the magnitude of posterior errors (Eq. 2) and some behaviour of the averaging kernels (Eq. 3) too. For the case shown, with 0.05° spacing, the mean posterior errors near the centre of the retrieval grid are $0.59\,\text{K}$ for SST and $0.46\,\text{m s}^{-1}$ for WSP, with corresponding DFS of 24.8 and 67.1. This imbalance in DFS indicates that there is more total signal for WSP retrieval, and thus an ideal setup for these two variables would be with a finer grid spacing for WSP and a coarser grid for SST; this was not tested here, but analysis of $\mathbf{A}$ could guide future algorithm development and selection of grid resolution for the chosen retrieval parameters. If the retrieval grid spacing is increased to 0.10°, the posterior errors for WSP decrease slightly to $0.44\,\text{m s}^{-1}$ but its DFS also decreases to 57.0, while those for SST are almost unchanged. Thus for a coarser retrieval grid, some information on WSP is wasted despite retrieved uncertainty decreasing due to lower smoothing error, whereas it is more optimal for SST.

Results from this synthetic experiment with the 2D-Var retrieval indicate that convergence can be achieved in two or three iterations, with fits to $T_B$s that follow the prescribed errors. Retrievals of SST and WSP are well constrained in the area of dense observations. There is minimal influence of a priori constraints on the retrieval's behaviour within the observation area due to the high density of the observations and their overlapping information content, with elements of $A$ summing to near 1 for all retrieval grid points within the observation area. However, for anomalies smaller than the retrieval's spatial sensitivity seen in Fig. 4, the prior still exerts some influence. This can be seen in retrieved versus true SSTs in Fig. 3, where positive and negative anomalies in left centre of the SST field are not well resolved by the 2D-Var because they are slightly smaller than the achieved resolution of the retrieval. In contrast, the achieved spatial resolution for WSP is much finer due to most information coming from the higher resolution channels.

## 5 Observed scenes results

In this section two observed ocean scenes from September 21, 2016 are examined. These scenes were chosen because they contain significant SST and WSP gradients and featured no detectable cloud liquid water. The first case is from an ascending orbit in the tropical Atlantic near the West African coast in the early afternoon. The second case is from a descending orbit, south of Australia in the Southern Ocean in the middle of the night. The first case has high SSTs and low WSP, and is also of interest because it is near the edge of the AMSR2 scan with pixels close together, whereas the second case features high WSP and cold SST. Both scenes were screened for cloud liquid water using the Remote Sensing Systems (RSS) AMSR2 v8 0.25° gridded product (Hilburn and Wentz, 2008) prior to selection.

The focus in this section is on the match to observed $T_B$s and attendant retrieval diagnostics, rather than comparison of retrieved parameters with other products, as the retrieval results and sensor calibration have not been validated or tuned. Validation or quantitative comparison with values from models or other retrievals is outside the scope of this study. Instead, this section examines two distinct scenes to probe the feasibility and flexibility of the 2D-Var approach with real observational data.

The 2D-Var settings for the observed cases are quite similar to those of the synthetic retrievals described earlier, with the same 0.05° retrieval grid spacing. The a priori errors are increased to 2.0 in the respective units for both SST and WSP but with

the same decorrelation length of $1.0°$, so as to limit influence of the prior and not over-constrain the retrieval. The a priori SST and WSP fields are from ERA5 and interpolated to the retrieval grid. Observation errors are increased to double the NEDT values along the diagonal of $\mathbf{S_y}$ to compensate somewhat for the existence of forward model and parameter errors. Also, for reasons that will become clear, $\mathbf{S_y}$ diagonal terms were adjusted for the second case presented, from twice NEDT to $50\,\mathrm{K}$ for

the 6H/7H/10H channels, to effectively remove all weighting from these channels. All information on the geometry for the forward model comes from the JAXA level 1 data.

## 5.1   Matching observed $T_B$s

The match to observed $T_B$s for the first case is shown in Fig. 6. As with the match between observed and simulated $T_B$s in Fig. 5, the dots indicate observation centres and $T_B$ differences are shown for all 12 AMSR2 channels used in the 2D-Var.

The observation locations come from the level 1 data, and the increased density of pixels near the edge of scan (i.e. the points located further south and west) is visible.

    Magnitudes of the $T_B$ differences are striking in Fig. 6, with low frequency channels matching the observations within $0.5\,\mathrm{K}$ for all pixels used in the 2D-Var. The match is not much worse for the higher frequency channels either, with maximum $T_B$ differences of $3\,\mathrm{K}$. Comparing these values with the specification sensor noise values for AMSR2 in Table 1, this is a

remarkably tight fit to the observed observation vector as the fit for many channels is better than may be expected due to sensor noise alone. It is noted that on-orbit performance for AMSR2 has indicated lower than specification noise levels for all channels (Kasahara et al., 2015, Fig. 4). Regardless, given the existence of forward model errors and calibration errors as well as sensor noise, it is surprising just how well the retrieval can match observed $T_B$s in this scene. Despite the tight SST gradient from southwest to northeast and their large FOVs, the forward model simulates all low frequency $T_B$s to a precision better than was

expected even if NEDT were the only observation error source.

    Comparison of retrieved fields with the a priori fields from ERA5 is shown in Fig. 7, with ERA5 on the left and the 2D-Var on the right. Due to the uncalibrated nature of the 2D-Var, results here are presented as anomalies from the mean field to make comparison with other datasets more instructive. Retrieved values mostly fall within the assumed errors of $2\,\mathrm{K}$ and $2\,\mathrm{m\,s^{-1}}$ with respect to ERA5 values, and demonstrate some effects at the edge of the observation space caused by the a priori spatial

correlation imposed. The strong gradient in SST seen by ERA5 is largely replicated by the 2D-Var, but the retrieved results show a stronger gradient in SST in the top half of the observation space. The retrieval finds more variability in WSP than found in ERA5, with higher WSP values found on the northern and southern ends of the observation area. Lower SST values found in the centre of the retrieval grid appear to be driven by behaviour of the 6H/7H channels, which show small systematic negative biases in Fig. 6 that could signal a calibration issue. By the same token, the higher WSP values from the retrieval are driven

partly by the positive observed minus simulated $T_B$s seen at 89H. ERA5 assimilates six channels of AMSR2 data from 18.6 to $89\,\mathrm{GHz}$, so these are not entirely independent estimates; however, due to the limited AMSR2 data points used by ERA5 within the assimilation window for the two scenes (not shown), there is little overlap of the total information content in these scenes. It is worth noting that ERA5 does not assimilate the 6H, 7H, or 89H channels from AMSR2.

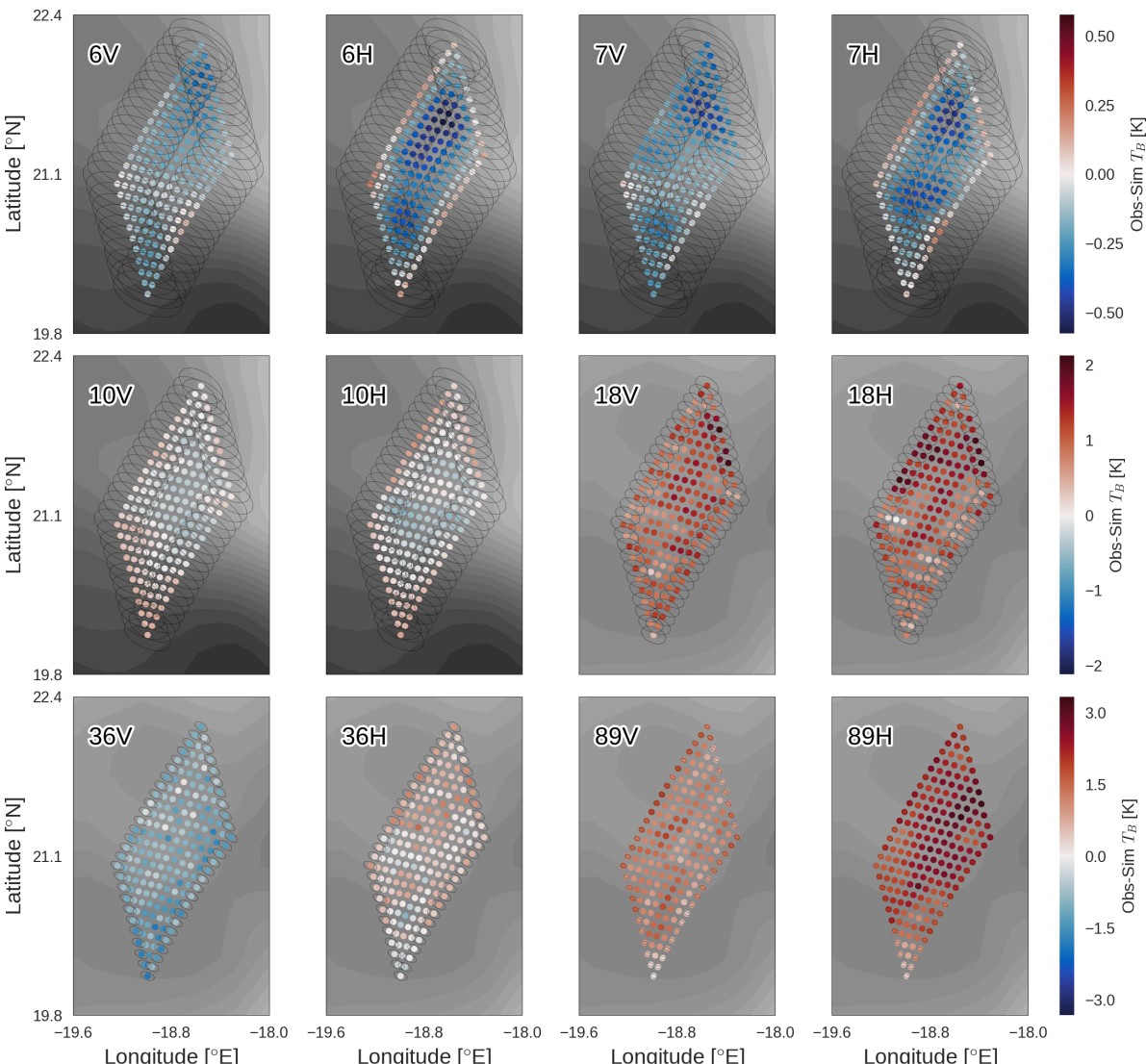

**Figure 6.** Observed minus simulated $T_B$ for 12 AMSR2 channels after completion of the 2D-Var retrieval, near the coast of Mauritania in the Tropical Atlantic on September 21, 2016. The background shows SST in the first six panels and WSP in the last six, with darker colours indicating higher values. Pixels on the edge of the measurement space are plotted along with their respective FOVs.

Whereas ERA5 is a reanalysis product, Fig. 7 also shows the gridded data from the widely used 0.25° RSS product (Hilburn and Wentz, 2008), which is a standalone retrieval from AMSR2. The anomaly patterns seen in SST and WSP are more similar between RSS and the 2D-Var than with ERA5. This is clearest for WSP, where the middle of the observation area shows a thin strip of higher wind speeds that are absent in the ERA5 analysis. The 2D-Var retrieves stronger gradients from north to south in the top and bottom halves of the observation area, but the overall pattern for WSP is similar if a little smoother in the RSS

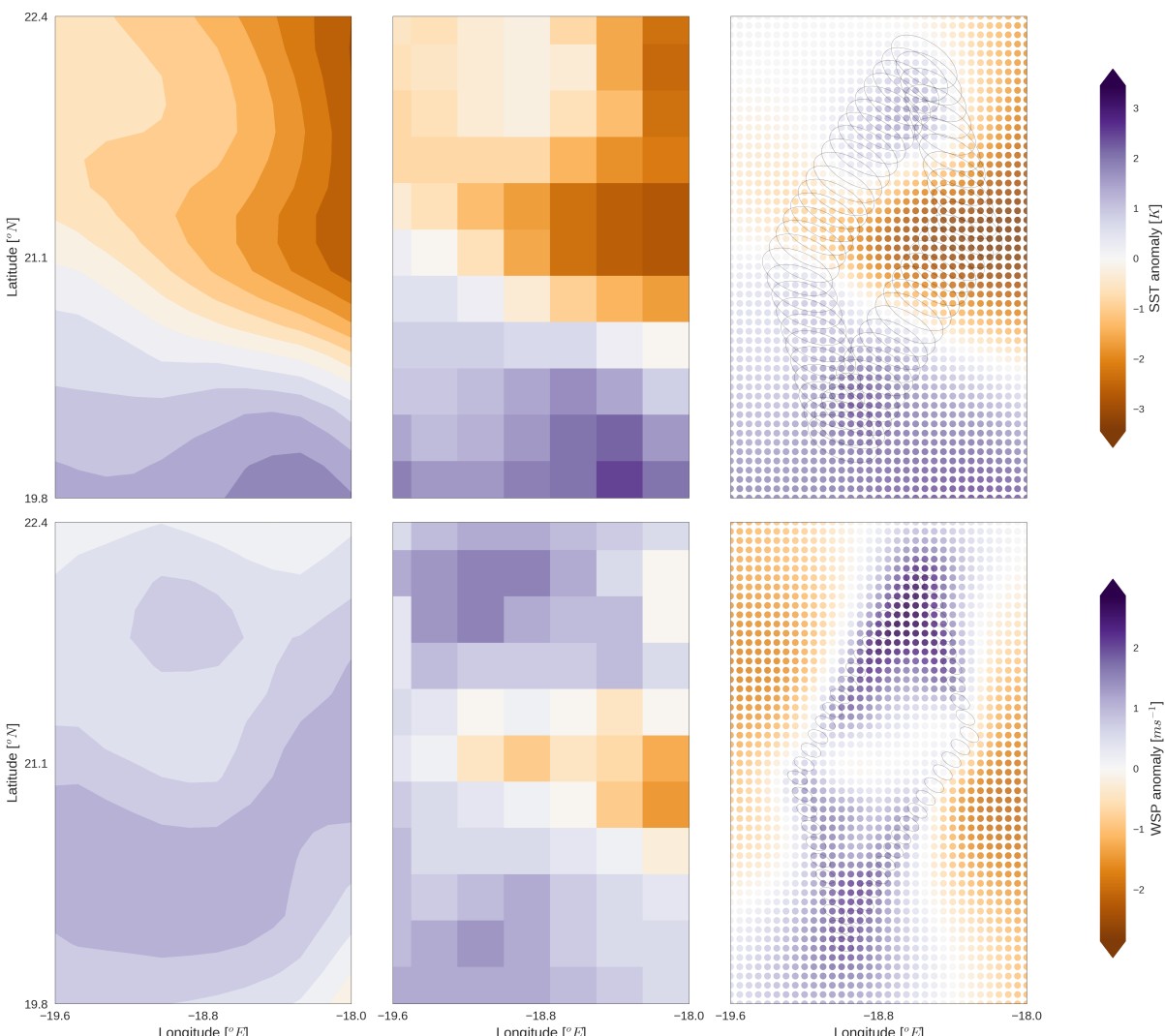

**Figure 7.** For the same scene as Fig. 6, the retrieved state vectors (right panels) are compared for SST (top row) and WSP (bottom row) against ERA5 (left) and RSS (middle). Each is displayed as the anomaly from the mean of the entire field. In the right panels, FOVs for the 6 and 36 GHz channels are shown for the pixels defining the edge of the observation area.

fields. For SST, the 2D-Var has a tighter and stronger gradient than RSS in the northern part of the observation area, but are otherwise in general agreement despite very different methodologies.

Fig. 8 follows the same style as Fig. 6 but for the Southern Ocean case. The different magnitudes of $T_B$ differences are notable in the colour scales, as very poor fits to observed $T_B$s were found at the low frequency H-pol channels of 6H, 7H, and 10H in the bottom left of the observation area. Initially the retrieval for this scene was run with the same $\mathbf{S_y}$ as the first case, but difficulties finding convergence due to the large errors at these channels caused the retrieval to be re-run with minimal

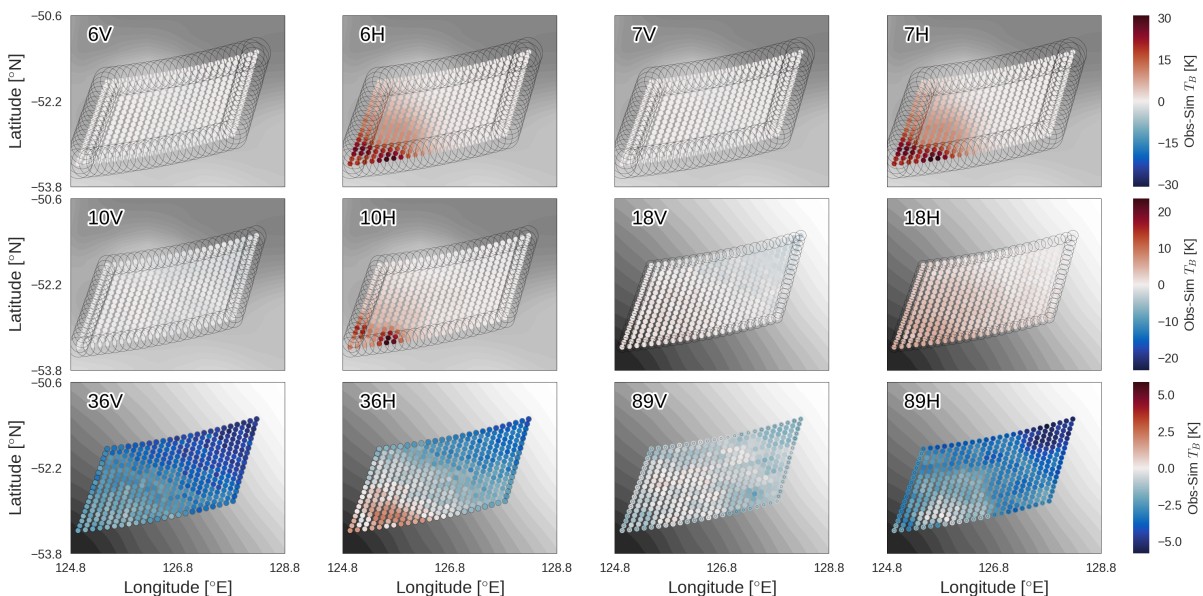

**Figure 8.** Observed minus simulated $T_B$ for 12 AMSR2 channels after completion of the 2D-Var retrieval, south of Australia in the Southern Ocean on September 21, 2016. The background shows SST in the first six panels and WSP in the last six, with darker colours indicating higher values. Pixels on the edge of the measurement space are plotted along with their respective FOVs.

weighting placed on 6H, 7H, and 10H. Through most of the observational domain, however, fits to the observations are within a few Kelvin and in line with the fits seen in Fig. 6. For instance, 89V exhibits fits within the prescribed error of double its NEDT value, and most of the other V-pol channels perform similarly. The very strong diagonal gradient in WSP is manifest in the fit to $T_B$s at numerous channels, hinting at emissivity model errors that are dependent on both frequency and polarisation.

It seems possible that FASTEM has a low bias for the low frequency H-pol channels in the conditions of high winds and low SST, but that is speculation. The 2D-Var yields useful information in this challenging Southern Ocean case, but it highlights the importance of the emissivity model and sensor calibration.

## 5.2 Retrieved spatial resolution

For the first observed case, the averaging kernels are examined to gauge the spatial resolution achieved by the 2D-Var. Shown
in Fig. 9 is the averaging kernel for a retrieval grid point near the grid's centre, following the style of Fig. 4. The spatial distribution of the averaging kernels, which appear elliptical and follow the angular orientation of the FOVs seen describing the observational space in Fig. 6, is distinctive when compared to Fig. 4. The lower panels of Fig. 9 show less symmetry than the synthetic case examined, though the cross sections are still relatively Gaussian at this grid spacing. The overlaid contour of the averaging kernel's HPBW, with three co-located AMSR2 FOVs for references, demonstrates that the achieved spatial
resolution for SST is approximately the size of the $10\,\mathrm{GHz}$ FOVs while for WSP it lies within the $18\,\mathrm{GHz}$ FOV.

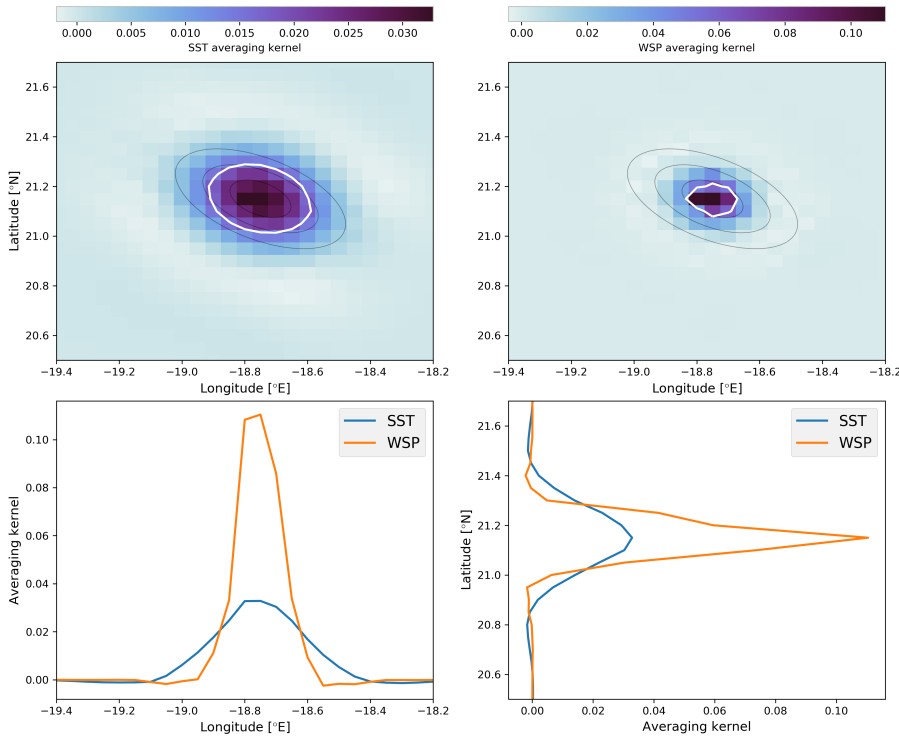

**Figure 9.** Averaging kernels from near the middle of the retrieval grid for the Tropical Atlantic case. The top panels show the 2D averaging kernels for SST and WSP for one point on the retrieval grid. Overlaid are outlines of the HPBWs of 6, 10, and 18GHz channels in black, along with a contour of the half-power of the averaging kernel in white. The bottom panels show longitudinal and latitudinal transects through the centres of the plots above.

The **A** values examined in Fig. 9 are not dissimilar from those seen for the synthetic case (Fig. 4). The orientation of the sensor's spatial response is in line with the elliptical FOVs of the observations, so it is unsurprising that **A** follows the antenna response of the channels with the greatest sensitivity for each retrieved parameter. In terms of kilometres, the 2D-Var achieves a retrieved spatial resolution of approximately $30\,\mathrm{km}$ for SST and closer to $10\,\mathrm{km}$ for WSP. Analysis using **A** carries the caveat
5   that not all terms are exactly known, but because these results are in line with those of the synthetic case that was constructed to yield exactly correct retrieval diagnostics, this gives confidence that Fig. 9 provides a reasonable estimate of achievable spatial resolution from a 2D-Var retrieval.

With respect to the posterior errors from the 2D-Var, those from this observed case are slightly larger than those from the synthetic case presented, as the assumed observation errors in $\mathbf{S_y}$ are larger. In the middle of the retrieval grid, mean posterior
10   errors for SST and WSP are $0.95\,\mathrm{K}$ and $0.80\,\mathrm{m\,s^{-1}}$, respectively, for the first observed case. Posterior errors are dominated by smoothing error (Eq. 4), accounting for about two thirds of the total error for both retrieved parameters. For this case the total

**Table 2.** Standard deviations (std) of observed ($T_{Bobs}$) and simulated brightness temperatures from the scene observed in Fig. 6, from the forward model applied to data from ERA5 ($T_{Bera5}$) and the retrieved state from the 2D-Var ($T_{B2dvar}$). Only selected channels are given. Approximate on-orbit NEDT values are from Kasahara et al. (2015).

| Channel | 6V | 6H | 10V | 18V | 36H | 89V | 89H |
|---|---|---|---|---|---|---|---|
| std($T_{Bobs}$) | 0.94 | 0.87 | 1.09 | 0.86 | 1.53 | 0.60 | 1.29 |
| std($T_{Bobs} - T_{Bera5}$) | 0.59 | 0.69 | 0.75 | 0.56 | 1.08 | 0.35 | 0.71 |
| std($T_{Bobs} - T_{B2dvar}$) | 0.09 | 0.19 | 0.23 | 0.31 | 0.43 | 0.34 | 0.54 |
| NEDT [K] | 0.34 | 0.34 | 0.70 | 0.70 | 0.70 | 1.20 | 1.20 |
| NEDT (on-orbit) [K] | 0.33 | 0.28 | 0.45 | 0.42 | 0.30 | 1.00 | 0.83 |

DFS was 77.6, of which 56.5 is for WSP and 21.1 for SST. This again demonstrates that an ideal retrieval grid spacing would thus be coarser for SST than for WSP, so as to reduce the large smoothing error seen here.

## 6 Discussion

### 6.1 Uncorrelated sensor noise

Sensor noise is assumed to be uncorrelated between channels in remote sensing applications, backed up by laboratory testing pre-launch. Potential spatial or temporal correlations of sensor noise are disregarded in standalone retrievals because 1D-Var retrievals are treated separately and justified in data assimilation through thinning procedures. In contrast, the treatment of overlapping FOVs in these 2D-Var case studies necessitates questioning this assumption. A quick glance at the $T_B$ fits seen in Figs. 5 and 6 shows a lack of noisy behaviour in the observed case, whereas the synthetic case was prescribed to exhibit channel noise as per the specifications for the AMSR2 sensor. The lack of noisy behaviour is also true for most channels in the second observed case (8), but the large emissivity model biases present caused this discussion to focus on the first case instead.

The very close match to observed $T_B$s exhibited in Fig. 6 was surprising. The observed minus simulated $T_B$ fields are smoother and less variable than those of the synthetic scene (Fig. 5) despite the prescribed noise being nominally the same. This would appear to have two possible explanations, namely over-fitting from the 2D-Var or sensor noise that is not truly independent from pixel to pixel. Due to the comparison with the synthetic scene, in which $T_B$ differences appear quite random after the 2D-Var has converged, the latter explanation seems more likely, i.e. sensor noise is not truly random and independent across or along scans from AMSR2.

Quantifying the fits to observations seen in Fig. 6, first we can analyse the standard deviation of observed $T_B$s from AMSR2 in that scene (Table 2). Spanning 11 scans of 17 pixels, the standard deviation of observed $T_B$ at 89V is $0.60\,\mathrm{K}$, less than the specified noise. This is despite the scene encompassing significant variability in SST and WSP according to ERA5, not to mention the sensitivity of 89V to water vapour variations. By itself this indicates some correlation in space or time of the sensor's noise characteristics, or overestimation of NEDT, and suggests that wholly uncorrelated noise for adjacent pixels is

a poor assumption. Even simulated $T_B$s from the a priori state (ERA5) are close enough to the observations to lie within specified NEDT levels for several channels. The standard deviations of observations compared to the retrieved 2D-Var state are remarkable across the spectral range of AMSR2, exhibiting fits that are a fraction of NEDT. Also given in Table 2 are approximate NEDT values estimated on-orbit by JAXA (Kasahara et al., 2015). These are notably lower than specification values at some frequencies, but fits are still well within these limits at several channels.

If the retrieval is essentially over-estimating the sensor noise, then the posterior errors reported by the 2D-Var could be over-estimates as well, and even tighter fits to observations may be possible. This was briefly tested, using prescribed errors just 20% of NEDT with the retrieval otherwise unchanged. In this experiment the 2D-Var was able to fit observations slightly closer at several low frequency channels, with slightly degraded fits at higher frequency H-pol channels. This indicates that even the extremely low noise values hinted at in Table 2 still overestimate the true sensor noise in the way it is traditionally represented, at least for this observed scene.

## 6.2 Spatial correlation and antenna patterns

To assess the 2D-Var results against more traditional microwave imager retrievals–like those derived from Backus-Gilbert convolution (Backus and Gilbert, 1967), 1D-Var, or regressions like RSS–is not straightforward. Backus-Gilbert requires a "target" resolution, and we have argued that a strength of the 2D-Var is indeed avoidance of specifying such a resolution target, allowing optimal retrieved resolution of multiple targets. The 2D-Var output is a gridded field by design, whereas a series of 1D-Var retrievals at each bore sight would need interpolation to a common grid. Furthermore, such comparisons have to be done for a synthetic scene as no absolute verification exists for the real scenes examined.

With these limitations in mind, experiments were devised to mimic 1D-Var retrievals and test the importance of a priori spatial correlations and the antenna pattern simulation in the results presented. First, the experiment to mimic 1D-Var in the synthetic scene neglected the antenna patterns in the forward model and all spatial correlation in the prior (diagonal $S_a$), with a single pencil beam forward model simulation at each bore sight. Retrieval grid resolution was decreased to $0.10°$ to ensure each grid box contained at least one bore sight. The results of this experiment are in Fig. 10 and may be contrasted against Fig. 3. Retrieved fields are less smooth for this experiment and exhibit bigger departures from the mean, as might have been expected.

Comparison of 2D-Var retrieval skill is given against the 1D-Var mimicking experiment in Table 3. Since these are at different retrieval grid spacing, the 2D-Var was run again at $0.10°$ resolution for a fairer comparison. Root mean squared error of retrieved parameters is clearly worse for the 1D-Var mimicking experiment, as is the correlation with truth values of both fields. The 2D-Var results at coarser spatial resolution prove that this is not a confounding factor here, as these particular metrics indicate an improvement in the 2D-Var retrieval skill instead.

Lastly, to pull apart the impacts of antenna pattern simulation and spatially correlated a priori constraints, a simple synthetic experiment was run. A step function jump in SST was defined at the centre of the synthetic grid, the same at all latitudes, with retrieval setup otherwise unchanged. Retrieval skill was then judged for differing spatial correlations assumed in $S_a$, including one retrieval neglecting the antenna pattern with zero spatial correlation. Results of this experiment are in Fig. 11. This sharp

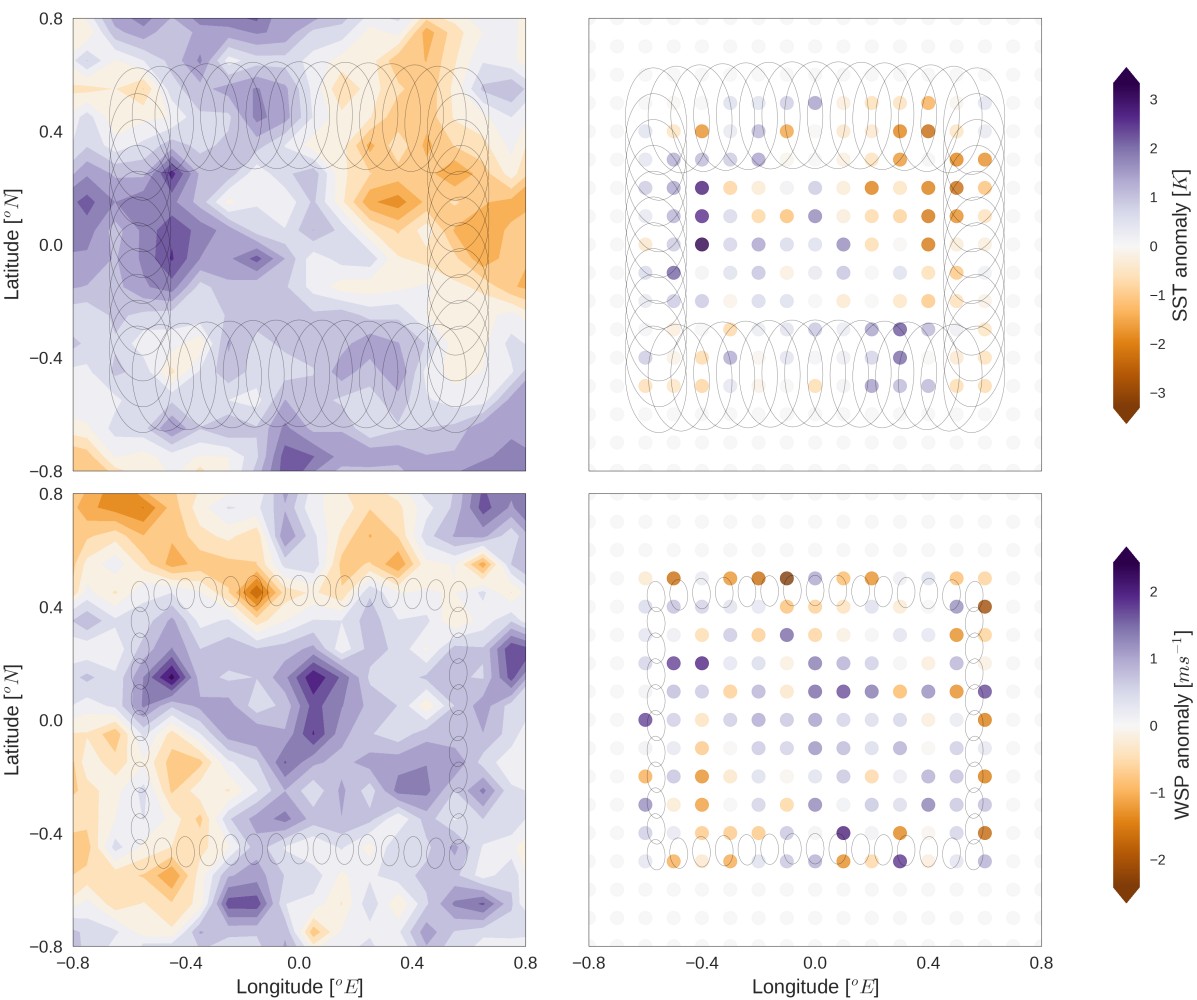

**Figure 10.** As with Fig. 3 but with no spatial correlations accounted for in $S_a$ and a "pencil beam" forward model at each bore sight (i.e. no accounting for antenna pattern) to approximate a series of 1D-Var retrievals.

edge in SST is smoothed out in all realisations, as expected given the large FOV size of channels most sensitive to SST and the dominance of smoothing error in the total posterior error seen earlier. With no spatial correlations assumed (akin to the 1D-Var mimicking experiment above), neglect of antenna pattern information causes greater spread in retrieved SST across latitudes. Accounting for antenna pattern does not help to better resolve the sharp edge on average, but it does significantly reduce the spread for all correlation lengths. The choice of decorrelation length ($\ell_d$) does not aid the retrieval in resolving the sharp gradient in SST. This finding is in line with the previous conclusion that the retrievals are well constrained by observations and not reliant on prior constraints within the observation area due to the high density of overlapping observations. This decrease in spread helps to explain the lower RMSE found by the 2D-Var in the previous experiments (Table 3).

**Table 3.** Root mean squared error (RMSE) and correlation coefficients (r) of retrieved versus true values for SST and WSP. Three different realisations of the retrieval are presented for the synthetic scene. Only grid points within the observation area are used in the calculations. The retrieval grid resolutions are given in parentheses–see text for details.

| | 2D-Var (.05°) | 2D-Var (.1°) | 1D-Var mimic (0.1°) |
|---|---|---|---|
| $RMSE_{SST}$ [K] | 0.36 | 0.35 | 0.94 |
| $RMSE_{WSP}$ [$ms^{-1}$] | 0.41 | 0.31 | 0.69 |
| $r_{SST}$ | 0.73 | 0.89 | 0.65 |
| $r_{WSP}$ | 0.79 | 0.81 | 0.59 |

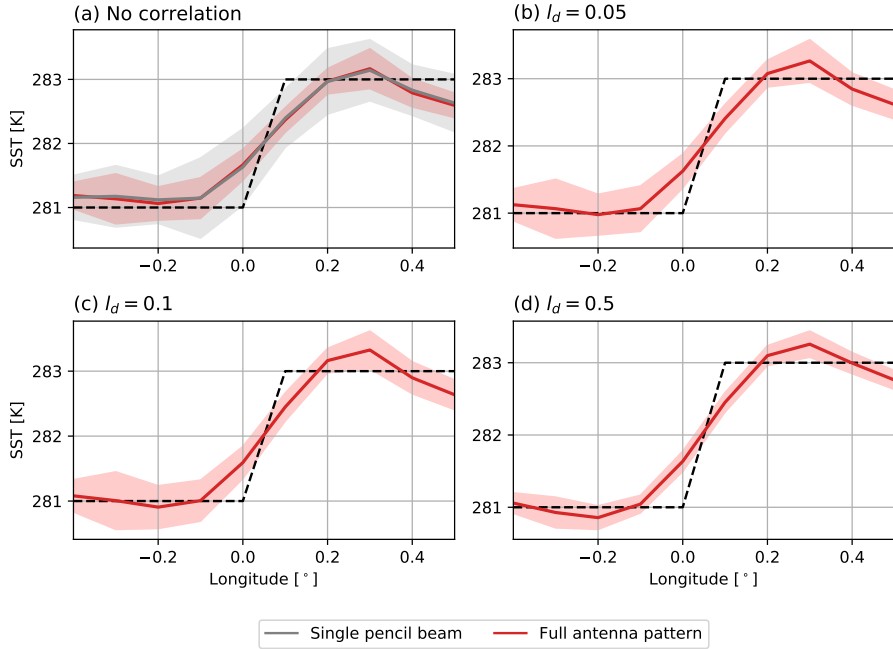

**Figure 11.** A synthetic scene as Fig. 3, but with the background defined by a step function in SST, shown by the dashed line and consistent across latitudes. The shaded areas represent the spread of retrieved SST in the latitudinal direction and the solid lines the means.

## 7 Summary and conclusions

In this study a 2D-Var retrieval has been presented, inverting radiances at a dozen microwave imager channels to solve for near-surface wind speed and skin temperature in oceanic scenes. To the authors' knowledge this is a novel application of the optimal estimation methodology, using explicit simulation of antenna beam patterns and spatial correlations on a fine retrieval grid to solve for an entire scene simultaneously. Whereas 2D-Var has been used in the context of scatterometer retrievals' ambiguity removal for some time (Hoffman et al., 2003; Vogelzang and Stoffelen, 2017) the retrieval presented is a standalone 2D-Var for

passive sensors. The use of 3D radiative transfer and sampling by pencil beam calculations in the ARTS-based forward model makes the 2D-Var retrieval possible, with the full geometry of the AMSR2 sensor taken into account. While computationally expensive, the 2D-Var inversions were not prohibitively so. Retrievals of two variables on a $0.05°$ grid spanning hundreds of kilometres with all channels' antenna patterns simulated with high spatial fidelity took on the order of ten minutes on a desktop, with no optimisation pursued. Larger inversions would be possible with the same setup.

A goal of this proof of concept study was to improve the spatial resolution of retrieved parameters by leveraging the spatial oversampling that is typical of low frequency microwave channels from space-borne radiometers. This goal was met, with the spatial response judged using the averaging kernel and compared against the size of different channels' FOVs in Figs. 4 and 9. For both the synthetic and real cases, the averaging kernels showed that the 2D-Var achieved spatial resolutions for SST on order of the 10GHz FOV, or about $30\,\mathrm{km}$. While this may not appear remarkable, most of the information content for the SST retrieval is derived from the 6 and 7GHz channels due to their greater sensitivity and lower NEDT values. For WSP the achieved spatial resolution was closer to that of the retrieval grid, smaller than the 18GHz FOV or about $10\,\mathrm{km}$. This is comparable to the real resolution achieved by current scatterometer retrievals (Vogelzang et al., 2017). Without the 2D-Var approach, the spatial resolution of a microwave imager retrieval is hard to judge and may be assumed to be that of the largest FOV of all channels utilised.

By examining the information content of the inversion via the degrees of freedom for signal, it was clear that the fine retrieval grid used in each case was suitable for WSP, but the lower DFS value for SST indicated that a coarser retrieval grid spacing would be optimal. Retrieval of both SST and WSP was well constrained by the observations, with the main influence of a priori information seen in its spatial correlations and constraint outside of the observation area in Figs. 3 and 7. The overlapping information content of the low frequency channels from AMSR2 provides significant redundancy if the calibration and forward model errors are well characterised, as demonstrated by the observed versus simulated $T_B$s in Figs. 6 and 8.

It has been argued that the methodology presented is a more optimal use of total information content from microwave imagers than offered by $T_B$ resampling and other retrieval methodologies. While limited in scope to cloud-free scenes in which a 2D-Var retrieval suffices, this methodology may be extended to 3D-Var standalone retrievals that encompass water vapour and hydrometeors. The 2D-Var can retrieve different parameters at the highest possible resolution permitted by the sensor's sensitivity, with different retrieval grid resolutions, with physically consistent treatment by the forward model, and no compromises necessary in terms of target resolution or increased noise.

Utilisation of all available spectral and spatial information content from current sensors is a worthy goal for retrievals and data assimilation schemes. The results presented here demonstrate that explicit forward modeling of antenna patterns can provide an alternative to the resampling methods common in the satellite retrieval literature, with no compromises necessary such as a target resolution or increasing the noise of some channels. Instead, the 2D-Var approach permits use of all channels at their native resolution while retrieved parameters are output at the highest possible spatial resolution given the information content available. The match to observed $T_B$s may also be useful in a data assimilation context, as simulation within the published noise values for the AMSR2 sensor indicates the utility of simulating the full antenna patterns with their spatial oversampling. The poorer observational fit seen in Fig. 8 demonstrates that forward model errors and calibration need to be

very well characterised for successful retrievals, and that an operational deployment of such a retrieval would require further development and analysis.

*Code availability.*  The code used for analysis is all available in the form of Jupyter notebooks via a Zenodo archive, found in the references. The ERA5 data used in the analysis constitute modified Copernicus Climate Change Service Information [2019]. RSS data are available

5  via remss.com. The ARTS model and variational solver are available via radiativetransfer.org. AMSR2 data are available from JAXA via suzaku.eorc.jaxa.jp/GCOM/.

*Author contributions.*  DD conceived of the study, performed most of the analysis, and wrote the manuscript. PE guided the development and SP provided technical help and discussions. The ARTS code upon which the study relies has been developed by many contributors over many years, but the multivariate optimal estimation solver written by SP deserves special mention.

10  *Competing interests.*  The authors declare that they have no conflict of interest.

*Acknowledgements.*  This study was funded with support from the Swedish National Space Agency, for which the authors are grateful. Thanks to the editor and two anonymous reviewers for helpful comments that refined the paper. Thanks as well for the data access freely provided for the ER5 and RSS products, and also to JAXA for making AMSR2 data publicly available.

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
