# Peer review of "An experimental 2D-Var retrieval using AMSR2"

_Atmospheric Measurement Techniques, 2019_

## Referee Comment (RC1) · Anonymous Referee #1 · 15 May 2019

**1   Overall**

This paper evaluates SST and wind speed retrieval from AMSR-2 data using 2DVAR. While AMSR-2 data have been operationally used for SST and wind retrieval, this is a novel application of 2DVAR to the AMSR-2 data.

This is an attractive technique since it avoids the Backus-Gilbert resampling (or averaging) that is used to ensure that $T_B$s from multiple channels share similar effective footprints (or fields of view (FOV)), moves from a pointwise to field-based retrieval in order to take into account the spatial correlation of the true underlying wind/temperature fields, and exploits the oversampled measurements (especially at lower frequencies) to minimize noise and maximize spatial resolution. Additionally, the retrieved output values directly contain error estimates as well as the effective retrieval "footprints".

[Figure]

The research to evaluate this method is performed well and has anticipated all of my concerns. The paper is laid out well, and has sufficient references to motivate the problem and describe the previous work. The results using simulated and actual data are instructive and illuminating. The figures are clear and the English usage is good. Overall, this paper is quite well done. I recommend it for publication.

**2  Specific comments**

I have a few comments.

- Several times on page 2 and later on, "pixel" is used to refer to the measurements. I think a better word would be "measurement" or "observation". One may form an image from the AMSR-2 $T_B$ values, but as shown later in the paper each $T_B$ measurement really is a average over an extended area weighted by the antenna footprint. "Pixel" usually connotes a square bounding box, which is a stricter definition than needed.

  However, I'm all right with how pixel is redefined later in section 3.2. I think that definition should happen before the first usage in the introduction or (my preference) just use "measurements" until then.

- In equation 4, why are there asterisks between the terms? At first I thought this was a Kronecker product instead of a matrix multiplication, but after some time I think it is just a "regular" matrix multiplication operator.

- Page 5, line 29: the decorrelation distance is set to 1 degree. This is probably fine for the scope of this paper, but for global or high latitude retrievals, this may yield odd results. The linear distance of a degree of longitude or latitude varies as a function of latitude. For the second study region in the Southern Ocean, the

zonal decorrelation length is about 70 km, whereas in the first study region it is about 105 km.

- Page 7, line 27: the spacecraft position is assumed constant over the course of a scan. How long does an AMSR-2 scan take? Or, by how much does AMSR-2 really move over one scan? I think the assumption of constant position would introduce some small error that is zero along one scan edge but increases along the scan. This helps simplify the calculations and is probably okay for the scope of this paper, but I think this should be addressed in future work.

- Page 15: possible errors due to the emissivity model. I concur with the speculation that the FASTEM model may be inaccurate at low SSTs. I am not very familiar with that particular model but I know that a few microwave emissivity models are available. I suspect that it would not be too difficult to swap out a different emissivity model. Unless it can be done quickly for this paper, I certainly recommend evaluating other models in future work.

**3  Technical corrections**

I have a few technical corrections.

- Table 1, and the colorbar and axis labels in Figure 3: the units (GHz, K, km, etc) should not be italicized

- Page 9, line 26: change "according to Table 1" to "according to the NEDT values from Table 1"

- Page 10, line 2: I think "WSP SI292" was meant to be "WSP is 292 K" or probably "WSP \SI{292}{\kelvin}" since it looks like the "siunitx" package is in use

- Figures 3, 5, and 7: overlaying the "dots" to show the error values on top of a background of the "true" values is a clever way to show both quantities but it does make it a little hard to read. Overlaying the FOVs is helpful but it does not help the problem. However, I don't have a good suggestion on how to present it instead.
* * *

---

## Referee Comment (RC2) · Anonymous Referee #2 · 13 Jul 2019

Paper Summary:

The authors develop a 2DVAR optimal estimation retrieval approach that entails explicit simulation of antenna beam pattern and use of spatial correlations on a fine retrieval grid to solve for an entire scene simultaneously. Such an approach is novel since a) deconvolution noise-introducing brightness temperature averaging methods (to achieve a constant FOV size) are not needed, and b) higher spatial resolution for the retrieved parameters are achieved due to the way overlapped satellite brightness temperature fields-of-view are accounted for. The paper is well written and was an interesting read.

Recommendation and General Comments:

I recommend major revisions.

The first reason for this recommendation is that I think this should be put into per-

spective with one of the other common retrieval approaches people take: 1. either neglecting the FOV differences in the optimal estimation, or 2. performing a Backus-Gilbert convolution/deconvolution to, perhaps, the middle-ground FOV size here (say corresponding to the ∼10 or 23 GHz channels here). For example, consider #1: suppose using your same forward model code, you did the retrieval one pixel at a time as the other papers cited here in the introduction do (1DVAR) – what would the surface wind, SST and simulated Tb fields look like for the same areas? Larger biases? Similar biases? Or, consider #2: if you did the B.-G. convolution/deconvolution, what would the results look like? The method here is novel, but it should be put into context with the other more common approaches so that we know what we may be gaining. What if we aren't really gaining that much for the additional expense? People widely use the Remote Sensing Systems (RSS) products at 0.25 resolution for wind and SST (which is comparable to the resolution here). Perhaps they don't go through the rigor and theory of establishing that they are really retrieving at 0.25 degrees – but, if it's close enough at a first-order level so that a more advanced retrieval achieves only second-order advances and second-order changes in biases – must we go there? For the September 21st case, maybe even RSS 0.25 deg. pixels look like what is shown here and they demonstrate similar wind and SST patterns? In such a case, a conclusion would be: "we can do what we propose here, but we can get high-res retrievals that are good enough with current passive microwave 1DVAR approaches."

So overall, we should at least know how what is done here compares to at least one other approach others often use. I realize the authors say they do not compare the retrieved parameters to other products since sensor calibration has not been validated, and so I am * not * asking them to do that here. What I am asking about is using their same forward model and retrieval code, and doing a quick 1DVAR and/or B.G.-first-then-retrieve approach. Such analysis could be added as additional figure panels adjacent to the ones already shown here for wind, SST and TBs so that we can visually compare.

Another major issue pertains to the figures. I had to stare at a number of the figures (Figs. 3, 5, 6, 7, 8) to absorb the information far longer than I do for most other papers I read. I'm wondering if it would be better to put the retrievals or simulated Tbs on separate panels, thus making a 4-panel images? It is tedious spending so much time to distinguish the fields and their heterogeneity and comparisons (and I'm still not even sure I see the contours well), perhaps so much so that it wipes out the advantages of having fewer panels per image.

A final smaller issue I am wondering about pertains to how to treat the edges of the grids. I am assuming this can't be run for an entire orbit at once. Does that mean there will be discontinuities arising near edges if this approach were executed or that distinct jumps at the seams of grids that are adjacent along the orbital track would be evident? What introduction of such artifacts that do not exist in 1DVAR be worth it to pursue a 2DVAR framework?

Specific Questions and Comments:

Sa specification: How sensitive is this to how you define off diagonal elements (spatial correlation lengths essentially)? So, this 2DVAR approach here allow us to not worry about deconvolution and how to treat overlapping FOVs in 1DVAR approaches, but then we have to newly account for spatial correlations (which must be very scene and atmosphere-feature dependent) in addition to the new issue of grid edges discussed above.

P4, line 30: I do not think 1DVAR must have a vertical dimension. A retrieval of parameters at the surface (e.g. X = [SST, surface wind, salinity]) using certain wavelengths would be a 3 parameter 1DVAR retrieval with no vertical dimension.

P5, Eq 4: This is matrix multiplication, right? Remove asterisks?

P13, lines 20-24: All of the dots look equally spaced in Figs. 5 and 6. I am not sure I understand the comment about increased density of pixels being clearly visible to the

S and W.

P13, line 25-32: I am not sure I agree with this text. And, visually, I do not see that the fits are different than expected from specifying NEdT in Sy (or, if I've read the text correctly, 2 X NEdT was specified in Sy, which would amount to 0.68K for the 6.9 GHz channels and 0.86K for the 7.3 GHz channels). The fits just follow a Gaussian with a width that is given by 2 X NEdT. Thus, 67% of the data will have fits that are smaller than 2 X NEdT (or less than 0.68K for 6.9 GHz). In other words, you should expect that many of retrievals have fits better than NEdT here (and 67% better than 2 X NEdT). I would consider it remarkable that all Tbs were matched if, simultaneously, the atmosphere is also in agreement with "truth". But, truth is not shown (or known). Retrieval algorithms are happy to swap goodness of the retrieved state (x) with goodness of the simulated radiances (and vice versa), 1DVAR included.

---

## Author Comment (AC1) · 22 Aug 2019

**General response**

Thanks to both reviewers for their careful readings of the manuscript. The following adjustments have been made to the manuscript in response to the reviewers' comments as well as the authors' own revisions. The manuscript revisions are still being conducted, so not everything is in its final form at present.

The authors decided, upon revision, that it is more appropriate to use "2D-Var" than "2DVAR" in the title and throughout the manuscript. Though this may seem trivial, the change should help to differentiate the 2D variational retrieval from the technique used in scatterometry (where 2DVAR is an acronym for 2D Variational Ambiguity Removal). This also brings the nomenclature in line with the lead author's current institution, ECMWF, which insists on "4D-Var" instead of "4DVAR" in all of its documentation, for instance. So while both variations are found in the literature, the authors decided to change all mentions of "xDVAR" to "xD-Var" in the revised manuscript for the sake of greater clarity.

In the following, relevant selections of the reviewers' comments are given in italics, with specific responses following each and any additional text added to the manuscript then given in quotes. Changes to the manuscript will be visible in the tracked changes document when that is submitted.

**Reviewer 1**

Specific Comments

*Several times on page 2 and later on, "pixel" is used to refer to the measurements. I think a better word would be "measurement" or "observation". One may form an image from the AMSR-2 TB values, but as shown later in the paper each TB measurement really is a average over an extended area weighted by the antenna footprint. "Pixel" usually connotes a square bounding box, which is a stricter definition than needed. However, I'm all right with how pixel is redefined later in section 3.2. I think that definition should happen before the first usage in the introduction or (my preference) just use "measurements" until then.*

All instances where "pixel" was used prior to Section 3.2 have now been changed. There are too many to detail them all here, but most have been changed to "observation centre" or "measurement," in line with the reviewer's suggestion.

*In equation 4, why are there asterisks between the terms?*

This was a typo and has now been fixed.

*Page 5, line 29: the decorrelation distance is set to 1 degree. This is probably fine for the scope of this paper, but for global or high latitude retrievals, this may yield odd results. The linear distance of a degree of longitude or latitude varies as a function of latitude. For the second study region in the Southern Ocean, the zonal decorrelation length is about 70 km, whereas in the first study region it is about 105 km.*

Yes, and indeed this effect is noticeable in the shape of the FOV ellipses for the Southern Ocean case, in which they more resemble circles due to the map projection. For this study it was simply easier to use a regular lat/lon grid and define the correlations relative to this, but of course conversion to kilometers using the appropriate distance formula could allow consistent treatment between scenes.

*Page 7, line 27: the spacecraft position is assumed constant over the course of a scan. How long does an AMSR-2 scan take? Or, by how much does AMSR-2 really move over one scan? I think the assumption of constant position would introduce some small error that is zero along one scan edge but increases along the scan. This helps simplify the calculations and is probably okay for the scope of this paper, but I think this should be addressed in future work.*

One AMSR2 observation has an integration time of 2.3, so for the purposes here this assumption should have little impact. To do a quick back of the envelope calculation, for the scenes considered of about 25 pixels across and 7km/s spacecraft velocity, the spacecraft moves about 450m. But the reviewer is right in that this does introduce a small error that would grow across the scan, which could become significant for very fine scales, wider swath retrievals, or the inclusion of cloud fields.

*Page 15: possible errors due to the emissivity model. I concur with the speculation that the FASTEM model may be inaccurate at low SSTs. I am not very familiar with that particular model but I know that a few microwave emissivity models are available. I suspect that it would not be too difficult to swap out a different emissivity model. Unless it can be done quickly for this paper, I certainly recommend evaluating other models in*

[Figure]

*future work.*

During the retrieval's development the authors considered use of the TESSEM2 model as well (which is already integrated into ARTS), but this showed larger residual biases than FASTEM at the stage of bias correction for the forward model, so we stuck with FASTEM. Many models also do not go below 10GHz or show poor performance at low frequencies. The other possible choice would have been the Meissner/Wentz model which has been specifically tuned for the C-band frequencies of AMSR-E and AMSR2, but because of proprietary concerns we were unable to integrate this model into the ARTS software.

Technical Corrections

*Table 1, and the colorbar and axis labels in Figure 3: the units (GHz, K, km, etc) should not be italicized*

Done.

*Page 9, line 26: change "according to Table 1" to "according to the NEDT values from Table 1"*

Done.

*Page 10, line 2: I think "WSP SI292" was meant to be "WSP is 292 K" or probably "WSP 292" since it looks like the "siunitx" package is in use*

Done (and you are absolutely right).

*Figures 3, 5, and 7: overlaying the "dots" to show the error values on top of a background of the "true" values is a clever way to show both quantities but it does make it a little hard to read. Overlaying the FOVs is helpful but it does not help the problem. However, I don't have a good suggestion on how to present it instead.*
This comment echoes one made by the other reviewer as well. In response the figures in the revised manuscript will be expanded to additional panels.

**Reviewer 2**

General comments

*...suppose using your same forward model code, you did the retrieval one pixel at a time as the other papers cited here in the introduction do (1DVAR) – what would the surface wind, SST and simulated Tb fields look like for the same areas? Larger biases? Similar biases? Or, consider 2: if you did the B.G. convolution/deconvolution, what would the results look like? ... People widely use the Remote Sensing Systems (RSS) products at 0.25 resolution for wind and SST (which is comparable to the resolution here). Perhaps they don't go through the rigor and theory of establishing that they are really retrieving at 0.25 degrees – but, if it's close enough at a first-order level so that a more advanced retrieval achieves only second-order advances and second-order changes in biases – must we go there? For the September 21st case, maybe even RSS 0.25 deg. pixels look like what is shown here and they demonstrate similar wind and SST patterns? So overall, we should at least know how what is done here compares to at least one other approach others often use.*

This comment generated a great deal of discussion amongst the authors. The final point (about comparison to the RSS gridded product) was anticipated by the authors and is an output of the plotting code (as well as ERA5 gridded fields seen in Fig. 7). These were not shown in the original manuscript due to concerns about the magnitudes of values shown, related to our concerns about the calibration of observations used. This comparison with RSS will be included in the revised manuscript either in the text or as a supplementary figure, possibly shown as anomalies from the mean state so

that this issue of magnitudes is not so important for interpretation.

To address the reviewer's comment, the authors considered doing separate 1D-Var retrievals using BG-style convolutions with the 2D-Var's averaging kernels defining the target size, though the treatment of sensor noise remains problematic and any choice of target resolution means running separate retrievals for each retrieval target since the sizes are different. In addition, comparison would have to be between the 2D-Var's gridded output fields and retrievals at the observation centres (possibly interpolated to a grid). Next, while the authors think that different versions of the real world retrievals (say using B.G. $T_B$s or separate 1D-Var retrievals) would be interesting, it would be difficult to draw concrete conclusions other than there being differences between the retrieved fields. Because of this, we have chosen to focus on the synthetic case since we know the true values in that case.

The fairest comparison, given our methodology and experimental setup, seems to be running the 2D-Var retrieval with zero spatial correlations and wholly non-overlapping FOVs so that the output retrieval grid is the same but grid points effectively do not talk to one another. In practice, this means that the true $T_B$ vector is calculated using the full antenna pattern in the forward model, but the retrieval forward model uses just a single pencil beam RT calculation at each boresight. The retrieval diagnostics will not be correct since the error assumptions and jacobians are not properly represented, but the variability of the retrieval's output fields can be compared directly since they are on the same grid and we know the true state. This comparison will be included in the manuscript's discussion section as a new subsection.

*I had to stare at a number of the figures (Figs. 3, 5, 6, 7, 8) to absorb the information far longer than I do for most other papers I read. I'm wondering if it would be better to put the retrievals or simulated Tbs on separate panels, thus making a 4-panel images?*

This echoes the final comment from reviewer 1. In response the figures in the revised manuscript will be expanded to additional panels.

*A final smaller issue I am wondering about pertains to how to treat the edges of the grids. I am assuming this can't be run for an entire orbit at once. Does that mean there will be discontinuities arising near edges if this approach were executed or that distinct jumps at the seams of grids that are adjacent along the orbital track would be evident? What introduction of such artifacts that do not exist in 1DVAR be worth it to pursue a 2DVAR framework?*

The authors think of the nesting of grids inherent with this approach in a similar manner to those of mesoscale models, for which there may be one or several outer grids at coarser resolution. Here the "observation area" of AMSR2 observations should lie wholly within the retrieval grid area, which all lies within the geophysical space of the 3D radiative transfer. While there may be strange behaviour at the edges of the retrieval grid where no observations exist and thus there is no sensitivity, these could be trimmed off before handing the data to users so that only parts of the retrieval grid with sensitivity from the measurements are output. So while it would be computationally expensive to do simultaneous retrievals for an entire orbit this should be technically possible, though some post processing would likely be helpful.

Specific Questions and Comments

*Sa specification: How sensitive is this to how you define off diagonal elements (spatial correlation lengths essentially)? So, this 2DVAR approach here allow us to not worry about deconvolution and how to treat overlapping FOVs in 1DVAR approaches, but then we have to newly account for spatial correlations (which must be very scene and atmosphere-feature dependent) in addition to the new issue of grid edges discussed above.*

It is a fair point that spatial correlations are indeed a new feature to worry about when running such retrievals. And as noted in discussion of the use of synthetic data, proper construction of the Sa matrix is important for interpretation of the retrieval's diagnostic

outputs. In the cases chosen, the decorrelation lengths assumed did not have much impact on retrieval results (not shown) except for at the edge of the observation area, where spatial correlation is essentially the only information – retrievals within the area of dense observations are well characterised by the measurement vector and a priori constraint is almost unnecessary. While the authors considered examination of this very issue for the synthetic case (defining the field with one decorrelation length but assuming a different one in the retrieval) this seemed secondary to the focus of the paper as a proof of concept, and since the subsequent real world examples do not represent a product for public consumption. Future work on 2D-Var retrievals should certainly consider their significance. As with a priori constraints used in all real world retrievals, future estimates of the correlation lengths can be informed by other data sets, model data, or even constant terms that are justifiable if the observations constrain the retrieval grid as tightly as the examples in this study.

*P4, line 30: I do not think 1DVAR must have a vertical dimension. A retrieval of parameters at the surface (e.g. X = [SST, surface wind, salinity]) using certain wavelengths would be a 3 parameter 1DVAR retrieval with no vertical dimension.*

This sentence has been modified accordingly: "In the context of passive microwave satellite retrievals, these are typically 1-dimensional variational retrievals, or 1D-Var ... as the state vector exists in one spatial dimension."

*P5, Eq 4: This is matrix multiplication, right? Remove asterisks?*

Done.

*P13, lines 20-24: All of the dots look equally spaced in Figs. 5 and 6. I am not sure I understand the comment about increased density of pixels being clearly visible to the S and W.*

It is admittedly not an overwhelming feature of Fig. 6, but the pixels are a little closer together on the ground at the edge of scans due to the conical scan pattern of AMSR3.

This sentence has been slightly revised from "clearly visible" to "visible."

*P13, line 25-32: I am not sure I agree with this text. And, visually, I do not see that the fits are different than expected from specifying NEdT in Sy...*

The revised manuscript will examine this quantitatively to determine whether the standard deviation of observed minus simulated $T_B$s differ significantly from the assumed NEDT values. This was an oversight in the original manuscript, so the authors appreciate this being questioned. The text referring to this will be amended accordingly.